# FineRMoE: Dimension Expansion for FineR-Grained Expert with Its Upcycling Approach

## Abstract

Fine-grained expert design has hitherto been restricted to the intermediate dimension of MoE layers, with its potential at the output dimension remaining largely unexplored, primarily due to the accompanying dimension discrepancy in subsequent computations after the MoE layers. Drawing on the power of multi-head attention, we pioneer the **FineRMoE** (FineR-grained MoE) architecture to expand fine-grained expert design across both intermediate and output dimensions for further enhancing expert specialization. FineRMoE introduces a bi-level sparsity paradigm: a sparse sum layer produces dimension-reduced candidate vectors for each token through its activated experts, and a sparse concatenation layer subsequently reassembles a dimension-restored output by selectively concatenating the chosen candidate vectors. Despite the bi-level sparsity, we devise a specialized routing mechanism that uses only a single router network to govern both expert activation and candidate selection, eliminating the extra computational cost of adopting two distinct routers. Meanwhile, to obviate the prohibitive cost of training FineRMoE from scratch, we adopt the upcycling paradigm for efficient expert construction and training. Nonetheless, existing upcycling methods are tailored to single-layer additive-fusion MoE architectures, and therefore not applicable to FineRMoE. To this end, we propose an upcycling method, which is compatible with prevailing ones, to accomplish FineRMoE in a cost-effective manner. By enabling flexible partition and expansion of pre-trained FFNs along the intermediate and output dimensions, the upcycling method promotes a broad adaptability in converting dense models into MoE models. Experimentally, we build the FineRMoE, in which 2 experts are sparsely activated out of 128 experts, based on Qwen2.5 with sizes of 0.5B, 1.5B and 7B via the proposed upcycling method. After continued training on 50B tokens, in comparison with baselines, FineRMoE exhibits superior performance across ten standard benchmarks, as well as remarkable efficiency in both parameters and inference. Extensive experiments validate the effectiveness of our FineRMoE architecture and upcycling method.

## 1 Introduction

Mixture-of-Experts (MoE) has emerged as the prevailing architecture of Large Language Models (LLMs) (Liu et al., 2024b; Team et al., 2025; Zeng et al., 2025; Chen et al., 2025; Comanici et al., 2025), yielding substantial gains across diverse tasks. By configuring the experts a markedly smaller intermediate size than that of the Feed-Forward Network (FFN), fine-grained expert design (Dai et al., 2024; Liu et al., 2024a) has been widely adopted for mitigating redundancy and improving specialization. Nevertheless, pushing fine-grained specialization by further shrinking either the input or output dimension of experts remains largely unexplored. The underlying factor is that such a design would disrupt the dimensional consistency of the computations before and after MoE layers.

Rethinking the multi-head attention (Vaswani et al., 2017), reducing the output dimension of QKV transformations drives heads toward distinct feature extraction, followed by the concatenation to reconstitute a richer representation. Motivated by this, reducing the output dimension of experts would encourage independent and low-rank representations, which in turn suppresses redundancy and enhances expert specialization. In addition, existing MoE models primarily rely on the weighted sum for multi-expert fusion, whereas concatenative fusion remains rarely exploited. Concatenation not

only rectifies the dimensional inconsistency caused by summing the compressed outputs of experts, but also expands the expert-combination space, thereby elevating the capacity of MoE models.

According to the preceding analysis, we propose the FineR-grained MoE (**FineRMoE**) architecture that generalizes fine-grained design beyond the intermediate dimension to the output dimension. FineRMoE comprises a shared expert and multiple finer-grained sparse experts. To achieve flexible adjustment of the granularity of sparse experts and model scale, we define four hyper-parameters for the joint architecture design, including intermediate granularity, intermediate expansion rate, output granularity, and output expansion rate. Regarding the forward process of the sparse experts in Fig. 1, we introduce a novel bi-level sparsity paradigm consisting of a sparse concatenation layer and a sparse sum layer. Specifically, for each token, its output with restored dimension from the sparse experts is obtained in the sparse concatenation layer by concatenating selected dimension-reduced vectors. In the sparse sum layer, each of these vectors is computed as the weighted sum of outputs from sparsely activated finer-grained experts within its corresponding MoE group.

Crucially, notwithstanding the sparsity of both the expert and vector levels inherent in the FineR-MoE, we forgo maintaining two distinct routers for each sparse layer. Instead, we design a novel routing mechanism that employs only a single router network to simultaneously trigger expert activation in the sparse sum layer and candidate vector selection in the sparse concatenation layer, thereby reducing parameter and computational cost associated with using two separate routers.

Despite the overwhelming advantages of MoE models, training them from scratch remains prohibitively expensive due to the extensive computational budgets and large-scale, high-quality data required. To facilitate the efficient construction and training of MoE models, the upcycling (Liao et al., 2025; Sukhbaatar et al., 2024; Jiang et al., 2025) paradigm has recently emerged. By leveraging a pre-trained dense LLM, upcycling converts the FFNs into MoE layers to avoid training experts from random initialization. Current upcycling methods are tailored to single-layer MoE models that fuse outputs from experts via weighted sum. They usually construct experts by duplicating the FFNs (Komatsuzaki et al., 2023; Zhang et al., 2024) or partitioning them along the intermediate dimension (Zhu et al., 2024; He et al., 2024). Consequently, these approaches are not applicable to our architecture with fine-grained design across both intermediate and output dimensions.

Based on the foregoing investigation, we contend that mainstream training-free upcycling methods can be unified under a single protocol, with the exception of methods requiring training for expert induction (Sukhbaatar et al., 2024; Zhang et al., 2024). To accomplish the proposed FineRMoE without training from scratch, we devise a novel upcycling method to instantiate finer-grained experts. By leveraging the four hyper-parameters defined in the FineRMoE, our upcycling method develops a configurable mechanism for expert construction. It enables flexible partitioning and expansion of the pre-trained FFN along both its intermediate and output dimensions, thereby rendering it applicable to both the proposed FineRMoE and conventional single-layer MoE architectures.

Experimentally, we build the FineRMoE based on the Qwen2.5 (Yang et al., 2024) with sizes of 0.5B, 1.5B, and 7B, by leveraging our upcyling method. After continued training on 50B tokens, the resulting models with 128 total experts and 2 activated achieve the best performance on ten benchmarks against curated baselines, and exhibit superior parameter and inference efficiency. Further ablation studies validate the effectiveness of the finer-grained design in mitigating redundancy and enhancing expert specialization, thus improving the performance. Our contributions are as below:

- **We propose the FineRMoE architecture.** To our best knowledge, it is the first to go beyond fine-grained expert design from intermediate dimension to the output dimension. It introduces a novel bi-level sparse paradigm consisting of a sparse concatenation layer and a sparse sum layer to process the input tokens following a dimension reduction-then-restoration order.
- **We introduce a specialized router mechanism.** Despite the inherent bi-level sparsity in the FineRMoE, the routing mechanism employs only a single router network to govern the activation in both sparse layers, thereby eliminating the need for two separate routers.
- **We develop a generalized upcycling method.** To build the FineRMoE in a cost-effective manner, the method enables efficient expert construction by flexibly partitioning and expanding the pre-trained FFN along both intermediate and output dimensions. With four adjustable parameters, it is generally applicable to both FineRMoE and conventional MoE architectures.
- **We provide extensive validation experiments.** Our FineRMoE architecture and the upcycling method achieve consistent performance gains across ten benchmarks, along with remarkable ef-

ficiency on both parameters and inference. Ablation studies systematically validate the effectiveness of FineRMoE in enhancing expert specialization and improving performance.

## 2 RELATED WORK

**Mixture-of-Experts (MoE).** It was initially proposed (Jacobs et al., 1991; Jordan & Jacobs, 1994) to scale model capacity while curb computational overhead (Masoudnia & Ebrahimpour, 2014; Chi et al., 2022), rendering it a prevalent building block in contemporary LLMs (Tang et al., 2025; Wei et al., 2024; Xue et al., 2024; Meta, 2025; Wang et al., 2025). Considering the granularity of experts, earlier MoE models (Du et al., 2022; Jiang et al., 2024) favor larger intermediate dimension to bolster per-expert capacity. Whereas LLMs (Yang et al., 2025; Liu et al., 2024b; Team et al., 2025) released recently have embraced fine-grained experts (Ludziejewski et al., 2024; Boix-Adsera & Rigollet, 2025) with lower intermediate dimension, which reduces redundancy and improves expert specialization (Dai et al., 2024). Nonetheless, existing fine-grained MoE models confine this design to the intermediate dimension. Motivated by the multi-head attention (MHA) mechanism, we aim to extend it to the output dimension of each expert for further specialization. Similarly, MH-MoE (Wu et al., 2024) is inspired by MHA to enhance granular understanding but emphasizes token partitioning, neglecting fine-grained expert design even in the intermediate dimension. In contrast, we devote our attention to the fine-grained expert design regardless of token splitting.

**Upcycling.** To circumvent the prohibitive computational and data demands of training MoE models from scratch, upcycling methods (Nakamura et al., 2025; Muennighoff et al., 2024) have recently emerged. To develop domain-specific experts, BAM (Zhang et al., 2024) and BTX (Sukhbaatar et al., 2024) first train separate copies of the pre-trained model on corresponding datasets, and then aggregate the resulting FFN weights into distinct experts to form the final MoE layers. Apart from these two methods, the vast majority of upcycling techniques instantiate experts via training-free strategies. One line of methods (Komatsuzaki et al., 2023; Vavre et al., 2024) initialize experts by replicating the pre-trained FFNs, while another line of work (Zhu et al., 2024; He et al., 2024) partitions the FFNs along the intermediate dimension to yield multiple fine-grained experts. Current upcycling approaches fall short of supporting the proposed FineRMoE architecture. To bridge this gap, we propose an upcycling method that enables the construction of FineRMoE, while remaining fully compatible with the two types of prevalent training-free expert construction methods.

## 3 METHOD

### 3.1 FINERMOE ARCHITECTURE

As depicted in the Fig. 1 left, the FineRMoE architecture consists of the shared expert and the sparse finer-grained experts, outputs of which are summed directly for later process. Each expert from the two types includes 3 weight matrices: the up projection weight $\boldsymbol{W}_1$, the gate weight $\boldsymbol{W}_g$, and the down projection weight $\boldsymbol{W}_2$. Given the LLM with hidden dimension as $h$ and an input $\boldsymbol{x} \in \mathbb{R}^h$, the output $\boldsymbol{y} \in \mathbb{R}^h$ of each expert is calculated as:

$$\boldsymbol{y}_i = \boldsymbol{x}\boldsymbol{W}_1 \odot \mathrm{SiLU}(\boldsymbol{x}\boldsymbol{W}_g), \quad \boldsymbol{y} = \boldsymbol{y}_i \boldsymbol{W}_2. \tag{1}$$

Denoting the intermediate size of the shared expert as $H$, the shared expert is thus composed of $\boldsymbol{W}_1^s \in \mathbb{R}^{h \times H}$, $\boldsymbol{W}_g^s \in \mathbb{R}^{h \times H}$, and $\boldsymbol{W}_2^s \in \mathbb{R}^{H \times h}$.

Setting the shared expert as a reference, sparse finer-grained experts are materialized through 4 hyper-parameters. For clarity, we introduce them in a top-down manner as shown in the Fig. 1 right, which includes the sparse concatenation layer and the sparse sum layer.

In the **sparse concatenation layer**, we define the *output granularity* $G_O$ measuring how many times larger is the hidden dimension $h$ of the LLM as compared to the output dimension $h_e$ of the sparse experts, which is calculated as:

$$G_O = h/h_e. \tag{2}$$

The *output expansion rate* $R_O$ is defined as the number of candidate dimension-reduced vectors to be selected from for each concatenation component. Therefore, the output $\boldsymbol{O} \in \mathbb{R}^h$ of this layer is the concatenation of $G_O$ components $\boldsymbol{O}_i \in \mathbb{R}^{h_e}$, and each component is selected from $R_O$ candidate

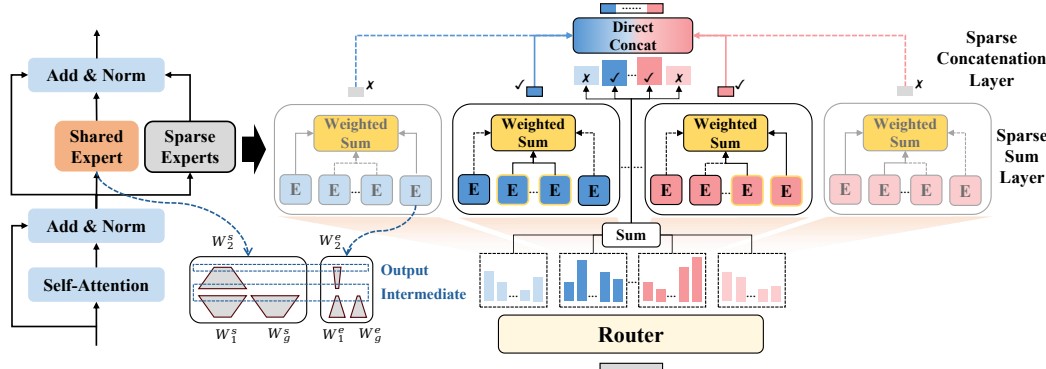

Figure 1: The proposed FineRMoE (FineR-grained MoE) architecture. The MoE layer is composed of the shared expert and multiple sparse experts, in which fine-grained design is applied to both the intermediate and output dimensions. The forward process of the sparse experts consists of a sparse concatenation layer and a sparse sum layer. A single router with specially designed routing mechanism simultaneously steers the activation in the two sparse layers.

vectors $\boldsymbol{O}_i^j \in \mathbb{R}^{h_e}$, which is the output of its corresponding group of experts in the sparse sum layer. Therefore, the concatenation process is formulated as:

$$\boldsymbol{O} = \text{Concat}(\boldsymbol{O}_0, ..., \boldsymbol{O}_i, ..., \boldsymbol{O}_{G_O-1}),$$
$$\boldsymbol{O}_i = \text{Top1Select}(\boldsymbol{O}_i^0, ...\boldsymbol{O}_i^j, ..., \boldsymbol{O}_i^{R_O-1}), \tag{3}$$
$$\boldsymbol{O}_i^j \in \mathbb{R}^{h_e}, \quad i \in [0, G_O - 1], j \in [0, R_O - 1].$$

The $\text{Top1Select}$ chooses the candidate vector with the highest corresponding router score as the concatenation component of the output, which will be detailed in Eq. 8 in the Sec. 3.2.

In the **sparse sum layer**, the experts are divided into multiple groups with each group consisting of $N_g$ finer-grained experts, the sparse weighted sum of which is the candidate vector $\boldsymbol{O}_i^j$ in the sparse concatenation layer. We define the *intermediate granularity* $G_I$ as measuring how many times larger is the intermediate size $H$ of the shared expert as compared to the intermediate size $H_e$ of sparse experts. The *intermediate expansion rate* $R_I$ is defined as how many times larger is the total sum of the intermediate size of sparse experts in a group as compared to the intermediate size of the shared expert. The definitions of these two hyper-parameters are as below:

$$G_I = H/H_e, \quad R_I = (N_g \cdot H_e)/H. \tag{4}$$

As each candidate vector in the sparse concatenation layer corresponds to a group of experts in the sparse sum layer, therefore, the total number of sparse experts $N$ is calculated as:

$$N = G_O \cdot R_O \cdot N_g = G_O \cdot R_O \cdot G_I \cdot R_I. \tag{5}$$

Each finer-grained expert $\boldsymbol{E}_k, k \in [0, N-1]$ is thus composed of the up projection weight $\boldsymbol{W}_{1k}^e \in \mathbb{R}^{h \times H_e}$, the gate weight $\boldsymbol{W}_{gk}^e \in \mathbb{R}^{h \times H_e}$, and the down projection weight $\boldsymbol{W}_{2k}^e \in \mathbb{R}^{H_e \times h_e}$.

According to the above, the candidate vector $\boldsymbol{O}_i^j$ in Eq. 3 is calculated as:

$$\boldsymbol{O}_i^j = \text{WeightedSum}(\boldsymbol{E}_k, \boldsymbol{x}), \quad i \in [0, G_O - 1], j \in [0, R_O - 1],$$
$$k \in [i \cdot G_I \cdot R_I \cdot R_O + j \cdot G_I \cdot R_I, \quad i \cdot G_I \cdot R_I \cdot R_O + (j+1) \cdot G_I \cdot R_I - 1]. \tag{6}$$

The $\text{WeightedSum}$ is conducted based on the router score corresponding to the sparse experts in each group, which will be detailed in Eq. 7 in Sec. 3.2.

## 3.2 ROUTER MECHANISM

Despite the intrinsic sparsity in the two layers of FineRMoE, we eschew the deployment of two distinct routers which will cause increased computation cost. Instead, as presented in Algorithm 1,

**Algorithm 1** Pseudocode of the Router Mechanism in a Pytorch-like style.

```
 1: # x: input    L: number of tokens in the input    N: number of experts
 2: # G_O: output granularity    R_O: output expansion rate
 3: # G_I: intermediate granularity    R_I: intermediate expansion rate
 4: # T_I: number of activated experts in each group in the sparse sum layer
 5: # Calculate the router score
 6: score = Router(x)                                           # L × N
 7: # Calculate the expert mask concerning the expert activation within each group in the sparse sum layer
 8: group_score = score.view(L, G_O R_O, G_I R_I)               # L × G_O R_O × G_I R_I
 9: _, group_act = TopK(group_score, T_I, dim=-1)               # L × G_O R_O × T_I
10: sum_mask = zeros(L, G_O R_O, G_I R_I)                       # L × G_O R_O × G_I R_I
11: sum_mask.scatter(-1, group_act, True)                      # L × G_O R_O × G_I R_I
12: # Calculate the expert mask concerning the candidate vector selection in the sparse concatenation layer
13: cc_score = sum(group_score, dim=-1)                        # L × G_O R_O
14: cc_score = cc_score.view(L, G_O, -1)                       # L × G_O × R_O
15: cc_act = argmax(cc_score, dim=-1, keepdim=True)            # L × G_O × 1
16: cc_mask = zeros(L, G_O, R_O).scatter(-1, cc_act, True)     # L × G_O × R_O
17: cc_mask = cc_mask.view(L, -1).unsqueeze(-1)                # L × G_O R_O × 1
18: cc_mask = cc_mask.repeat(1,1,G_I R_I)                      # L × G_O R_O × G_I R_I
19: # Element-wise AND operation on the two masks
20: final_mask = sum_mask & cc_mask                            # L × G_O R_O × G_I R_I
21: final_mask = final_mask.view(L, -1)                        # L × N
22: # Calculate the final scores and indices of activated experts
23: final_score = score.masked_fill(~final_mask.bool(), float(-inf))
24: probs, indices = TopK(final_score, G_O T_I, dim=-1)        # L × G_O T_I
```

the mechanism with only a single router is devised to simultaneously select the dimension-reduced vectors in the sparse concatenation layer and activate the experts within each group in the sparse sum layer. Specifically, after calculating the initial `score` by the single router in Line 5–6, the mechanism computes the activation mask over all sparse experts from two perspectives.

The first perspective corresponds to Line 7–11 computes the mask for expert activation within each group in the sparse sum layer. By dividing the initial `score` into $G_O R_O$ groups with each group containing $G_I R_I$ elements, $T_I$ experts with higher scores will be activated for the weighed sum and produce the candidate vector. Therefore, by considering only within a group of experts with indices $l$ ranging from $0$ to $G_I R_I - 1$, the WeightedSum in Eq. 6 is calculated as:

$$\text{WeightedSum}(\boldsymbol{E}_l, \boldsymbol{x}) = \sum_{l \in \text{ActSet}} \boldsymbol{E}_l(\boldsymbol{x}) \cdot \texttt{group\_score}[\text{any}, \text{any}, l],$$
$$\text{ActSet} = \texttt{group\_act}[\text{any}, \text{any}, :], \tag{7}$$

where any refers to any position of tokens and any group of experts.

The other perspective corresponds to Line 12–18 computes the mask concerning the selected vectors in the sparse concatenation layer. In detail, each concatenation component is selected from $R_O$ candidate vectors. The Top1Select in Eq. 3 chooses the candidate with the highest `cc_score`, which is the sum of scores of all experts in the corresponding group, and it is formulated as:

$$\text{Top1Select}(\boldsymbol{O}_i^j) = \boldsymbol{O}_i^{\texttt{cc\_act}[\text{any}, i, 0]}, \tag{8}$$

where any refers to any position of tokens. After that, the mask is then broadcast to all experts, by which the group of experts corresponding to the selected vector are not masked, and vice versa.

The `final_mask` is obtained by the element-wise AND operation between these two masks in Line 19–21. After applying the `final_mask` on the initial `score` and thus obtaining the `final_score`, $G_O T_I$ experts with higher score are activated for computation, as in Line 22–24.

Based on the router mechanism design, the computation process of a sequence of tokens in the sparse experts of FineRMoE is shown in Fig. 3 in the Appendix A. After the router assigns each token to its activated experts, the tokens are permuted and dispatched to the corresponding experts for parallel forward computation. Upon completion, they are unpermuted to restore the original sequence order. Within each expert group, the outputs pertaining to the same token are aggregated into a dimension-reduced vector via weighted sum. Then, the sparsely selected vectors are directly concatenated to yield the final outputs of the sparse experts.

### 3.3 UPCYCLING FOR FINERMOE

Training MoEs from scratch is notoriously expensive. As an efficient paradigm of training MoEs, existing upcycling methods are tailored to single-layer, weighted-sum MoEs, rendering them inapplicable to the proposed FineRMoE architecture. To this end, we present an upcycling method for training the FineRMoE efficiently. Given a pre-trained dense LLM with the FFNs composed of the up projection weight $W_1^p \in \mathbb{R}^{h \times H}$, the gate weight $W_g^p \in \mathbb{R}^{h \times H}$, and the down projection weight $W_2^p \in \mathbb{R}^{H \times h}$, the shared expert in FineRMoE is initialized by copying the pre-trained FFNs:

$$W_1^s = W_1^p, \quad W_g^s = W_g^p, \quad W_2^s = W_2^p. \tag{9}$$

As for the sparse finer-grained experts with index $k$ ranging from 0 to $N - 1$, their weights $W_{1k}^e \in \mathbb{R}^{h \times H_e}$ and $W_{gk}^e \in \mathbb{R}^{h \times H_e}$ are constructed by splitting the $W_1^p$ and $W_g^p$ along the intermediate dimension, while the weight $W_{2k}^e \in \mathbb{R}^{H_e \times h_e}$ is constructed by splitting the $W_2^p$ along both the intermediate and output dimensions. The detailed expert construction is formulated as:

$$W_{1k}^e = W_1^p[:, ((k \bmod (G_I R_I)) \bmod G_I) \cdot \frac{H}{G_I} : (((k \bmod (G_I R_I)) \bmod G_I) + 1) \cdot \frac{H}{G_I}],$$

$$W_{gk}^e = W_g^p[:, ((k \bmod (G_I R_I)) \bmod G_I) \cdot \frac{H}{G_I} : (((k \bmod (G_I R_I)) \bmod G_I) + 1) \cdot \frac{H}{G_I}],$$

$$W_{2k}^e = W_2^p[((k \bmod (G_I R_I)) \bmod G_I) \cdot \frac{H}{G_I} : (((k \bmod (G_I R_I)) \bmod G_I) + 1) \cdot \frac{H}{G_I},$$

$$\lfloor \frac{k}{R_O G_I R_I} \rfloor \cdot \frac{h}{G_O} : (\lfloor \frac{k}{R_O G_I R_I} \rfloor + 1) \cdot \frac{h}{G_O}], \quad k \in [0, N - 1],$$

$$\tag{10}$$

where mod is the modulo operation, and $\lfloor \cdot \rfloor$ is the operation that calculates the integer part of the division. Therefore, by configuring the 4 hyper-parameters $G_I, R_I, G_O, R_O$, the proposed upcycling method is not only limited to the FineRMoE architecture, but can extend to existing ones.

Specifically, by setting $G_O, R_O, G_I = 1$ and $R_I$ as the duplication times, upcycling by replicating the FFNs (Komatsuzaki et al., 2023; Vavre et al., 2024) can be implemented. By setting $G_O, R_O, R_I = 1$ and $G_I$ as the splitting times, upcycling via partitioning the FFNs along the intermediate dimension (Zhu et al., 2024; He et al., 2024) can be implemented.

## 4 EXPERIMENTS

We first compare the proposed FineRMoE trained via the proposed upcycling method based on Qwen2.5 (Yang et al., 2024) against baselines in Sec. 4.1. Then, Sec. 4.2 validates the effectiveness of the finer-grained design. Next, Sec. 4.3 presents the ablation study on the architecture of the FineRMoE. Besides, a series of experiments by configuring the 4 hyper-parameters differently are delivered in Sec. 4.4. Furthermore, the ablation study on the router design is analyzed in Sec. 4.5. We provide the experimental setup including training and evaluation details in Appendix B, with supplemental analysis and ablation studies in Appendix C–F.

### 4.1 BASELINE COMPARISON

Based on Qwen2.5 (Yang et al., 2024) with sizes of 0.5B, 1.5B and 7B, the baselines are as below:

**PT**: The official pre-trained models. **CT**: Continued training the dense models directly.

**C32A2** (Komatsuzaki et al., 2023): Copying the pre-trained FFN for 32 times as experts and 2 of them are activated. We implement C32A2 by setting $G_I = 1, R_I = 32, G_O = 1, R_O = 1$.

**S16A4** (Zhu et al., 2024): Splitting the pre-trained FFN for 16 times as experts and 4 of them are activated. We implement S16A4 by setting $G_I = 16, R_I = 1, G_O = 1, R_O = 1$.

**DU** (Nakamura et al., 2025): Replicating pre-trained FFN for 8 times first, then re-initializing 50% of the parameters in each weight matrix in each expert, and 2 experts are activated.

**NVShard** (He et al., 2024): Splitting the pre-trained FFN for 8 times and replicating all split parts for 8 times, resulting in 64 experts in total and 8 of them are activated.

Table 1: The performance comparison of the proposed FineRMoE against baselines. #P/B: Total parameter size. #AP/B: Activated parameter size. Hell.: HellaSwag. Wino.: WinoGrande. ARCC.: ARC-Challenge. ARCE.: ARC-Easy. AGIE.: AGIEval.

| | #P/B | #AP/B | MMLU | BBH | Hell. | Wino. | ARCC. | ARCE. | AGIE. | MBPP | GSM8K | GPQA | AVG |
|---|---|---|---|---|---|---|---|---|---|---|---|---|---|
| | | | | | | *Base Model: Qwen2.5-0.5B* | | | | | | | |
| PT | 0.49 | 0.49 | 47.50 | 23.18 | 51.84 | 56.67 | 34.81 | 68.90 | 28.07 | 29.40 | 41.70 | 29.80 | 41.19 |
| CT | 0.49 | 0.49 | 46.18 | 32.04 | 52.71 | 57.46 | 35.58 | 68.06 | 27.31 | 28.00 | 38.51 | 27.23 | 41.31 |
| C32A2 | 10.36 | 0.95 | 44.52 | 31.12 | 51.82 | 55.56 | 34.84 | 67.42 | 28.31 | 33.80 | 37.32 | 28.12 | 41.28 |
| S16A4 | 0.63 | 0.40 | 26.04 | 12.95 | 30.45 | 51.70 | 22.95 | 46.46 | 25.47 | 0.00 | 1.90 | 23.88 | 24.18 |
| DU | 2.83 | 0.94 | 24.36 | 6.93 | 28.51 | 50.67 | 21.93 | 40.70 | 25.86 | 0.00 | 2.27 | 25.00 | 22.62 |
| NVShard | 2.83 | 0.63 | 39.49 | 27.98 | 49.44 | 55.56 | 34.56 | 65.40 | 26.79 | 16.80 | 26.84 | 28.57 | 37.14 |
| FineRMoE | 1.68 | 0.65 | 45.43 | 30.81 | 51.74 | 55.88 | 34.90 | 67.55 | 28.19 | 33.40 | 37.60 | 28.35 | **41.39** |
| | | | | | | *Base Model: Qwen2.5-1.5B* | | | | | | | |
| PT | 1.54 | 1.54 | 60.87 | 43.33 | 67.84 | 64.88 | 54.86 | 81.02 | 39.83 | 43.40 | 65.73 | 32.14 | 55.39 |
| CT | 1.54 | 1.54 | 60.70 | 45.15 | 68.68 | 65.43 | 53.41 | 80.72 | 38.91 | 41.20 | 66.11 | 31.47 | 55.18 |
| C32A2 | 37.62 | 2.94 | 59.35 | 46.78 | 68.68 | 65.30 | 52.90 | 80.01 | 43.26 | 50.60 | 65.73 | 32.37 | 56.50 |
| S16A4 | 1.78 | 0.91 | 25.06 | 23.41 | 35.06 | 50.38 | 25.34 | 50.38 | 25.91 | 1.00 | 3.03 | 22.32 | 26.32 |
| DU | 9.87 | 2.93 | 25.93 | 10.17 | 30.07 | 52.41 | 20.99 | 43.27 | 25.52 | 0.00 | 1.90 | 24.78 | 23.50 |
| NVShard | 9.88 | 1.78 | 56.96 | 41.13 | 66.72 | 63.30 | 49.49 | 77.82 | 36.78 | 33.80 | 64.67 | 31.03 | 52.14 |
| FineRMoE | 5.64 | 1.85 | 59.64 | 47.09 | 68.17 | 65.43 | 53.33 | 80.81 | 43.14 | 50.40 | 65.81 | 32.37 | **56.62** |
| | | | | | | *Base Model: Qwen2.5-7B* | | | | | | | |
| PT | 7.62 | 7.62 | 74.28 | 68.30 | 80.26 | 76.32 | 63.65 | 87.12 | 56.20 | 63.60 | 84.31 | 35.04 | 68.91 |
| CT | 7.62 | 7.62 | 72.97 | 69.59 | 80.47 | 75.53 | 63.65 | 87.33 | 52.67 | 64.00 | 85.37 | 34.60 | 68.62 |
| C32A2 | 184.42 | 13.33 | 74.60 | 70.16 | 80.38 | 77.03 | 63.31 | 86.32 | 55.91 | 71.60 | 83.62 | 36.28 | 69.92 |
| S16A4 | 7.62 | 3.34 | 35.79 | 31.76 | 47.04 | 56.59 | 34.64 | 60.98 | 28.94 | 17.20 | 21.30 | 23.44 | 35.77 |
| DU | 47.54 | 13.32 | 25.71 | 17.98 | 34.07 | 51.85 | 25.51 | 51.30 | 25.65 | 0.00 | 2.81 | 23.66 | 25.85 |
| NVShard | 47.55 | 7.63 | 69.81 | 66.67 | 79.05 | 74.27 | 59.81 | 84.97 | 49.36 | 53.20 | 83.62 | 31.03 | 65.18 |
| FineRMoE | 26.65 | 7.94 | 73.08 | 71.22 | 79.51 | 75.69 | 63.40 | 86.95 | 56.70 | 69.40 | 85.37 | 39.06 | **70.04** |

According to Sec. 4.4 analyzed later, the 4 hyper-parameters of FineRMoE are configured as: $G_I = 32$, $R_I = 1$, $G_O = 2$, $R_O = 2$, leading to 128 total experts, and $T_I = 1$, leading to $G_O T_I = 2$ activated experts. Experiments for baseline comparison are performed by training on 50B tokens.

As shown from the results in Table 1, our FineRMoE achieves the best average performance at each model size investigated. Notably, while continued training on the same data slightly degrades the performance of the dense models compared to the pre-trained version, FineRMoE produces substantial gains. For dense models, all parameters are activated during both training and inference, meaning that continual pre-training with new data affects the entire model. This often leads to catastrophic interference as new knowledge conflicts with previously acquired knowledge. While the sparse activation of MoE models enables the model to acquire new knowledge efficiently while preserving its pre-trained capabilities with fewer conflicts. This consistent improvement demonstrates that upcycling into FineRMoE is a more effective strategy for leveraging additional data and avoiding the performance degradation.

Although C32A2 constructs MoE models with more than 6 times of parameters than ours, FineRMoE still achieves better performance, indicating its high parameter efficiency and effective expert learning. In contrast, though S16A4 minimizes parameter overhead, its performance collapses, which may be caused by the lack of shared expert. Subsequent ablation study on the shared expert in the Sec. 4.3 reproduces analogous observations, evidencing that shared experts are essential for sparse fine-grained experts. Besides, Drop-Upcycling achieves the performance far inferior to that of FineRMoE. We infer the reason as the existence of part of re-initialized parameters. In the paper of Drop-Upcycling (Nakamura et al., 2025), experiments are performed by training for 500B tokens. For a fair comparison with other methods, we limit training to 50B tokens. Consequently, Drop-Upcycling begins with a higher training loss and converges more slowly. This demonstrates that FineRMoE also exhibits data-efficiency compared with Drop-Upcycing in building MoE models from dense models. In addition, FineRMoE achieves an average performance advantage of around 5 points than NVShard across three sizes, validating the effectiveness of our method.

Furthermore, Appendix C demonstrates that our FineRMoE also achieves the optimal inference efficiency with lower latency and the highest throughput. Notably, FineRMoE's TTFT (178.3 ms) is 281 times faster than that of C32A2 (50245.9 ms), and its throughput (27.3 tokens/s) is 136 times higher than that of C32A2 (0.2 tokens/s).

Table 2: The effectiveness validation of the finer-grained design in the proposed FineRMoE based on Qwen2.5-1.5B by traning on 10B tokens. FG: Fine-Grained. Dim: Dimension.

| Settings | Pre-Trained | No FG | Inter FG | Out FG | Inter&Out FG |
|---|---|---|---|---|---|
| Intermediate FG | - | ✗ | ✓ | ✗ | ✓ |
| Output FG | - | ✗ | ✗ | ✓ | ✓ |
| $G_I, R_I, G_O, R_O$ | - | 1,1,1,2 | 32,1,1,2 | 1,1,2,2 | 32,1,2,2 |
| Intermediate Dim | 8960 | 8960 | 280 | 8960 | 280 |
| Output Dim | 1536 | 1536 | 1536 | 768 | 768 |
| #A-Experts / #Experts | - / - | 1 / 2 | 1 / 64 | 2 / 4 | 2 / 128 |
| #A-Param / #Param (B) | 1.54 / 1.54 | 2.93 / 4.09 | 1.82 / 4.09 | 3.70 / 5.63 | 1.85 / 5.64 |
| MMLU | 60.87 | 51.13 | 56.25 | 59.52 | 59.30 |
| BBH | 43.33 | 34.37 | 41.18 | 45.49 | 45.97 |
| HellaSwag | 67.84 | 66.59 | 66.26 | 67.52 | 67.51 |
| WinoGrande | 64.88 | 59.91 | 63.54 | 65.90 | 65.27 |
| ARC-C | 54.86 | 49.83 | 51.02 | 52.39 | 53.41 |
| ARC-E | 81.02 | 78.28 | 78.96 | 80.26 | 80.35 |
| AGIEval | 39.83 | 34.28 | 39.99 | 42.24 | 41.81 |
| MBPP | 43.40 | 32.60 | 41.00 | 49.60 | 50.00 |
| GSM8K | 65.73 | 41.39 | 62.24 | 67.25 | 66.34 |
| GPQA | 32.14 | 29.69 | 30.13 | 31.03 | 32.14 |
| Average | 55.39 | 47.81 | 53.06 | 56.12 | **56.21** |

Our experiments confirm the efficacy of the proposed upcycling method. Moreover, they demonstrate that the FineRMoE architecture achieves a superior trade-off between parameter efficiency and performance, along with optimal inference efficiency, outperforming all baseline methods.

## 4.2 EFFECTIVENESS VALIDATION OF FINER-GRAINED DESIGN

To evaluate the effectiveness of the finer-grained design in the proposed FineRMoE architecture, we experiment based on the Qwen2.5-1.5B by training on 10B tokens for efficiency. Four settings are included: (1) no fine-grained design (**No FG**); (2) fine-grained design only on the intermediate dimension (**Inter FG**); (3) fine-grained design only on the output dimension (**Out FG**); and (4) fine-grained design on both intermediate and output dimensions (**Inter&Out FG**).

According to Table 2, the poorest performance arises when fine-grained design is removed from both intermediate and output dimensions. Introducing fine-grained design only on the intermediate dimension brings noticeable improvement, but the performance still remains below that of the pre-trained model. In comparison, applying fine-grained design only to the output dimension produces a substantial gain by an average of 0.73 compared with the pre-trained model. This output-only fine-grained design outperforms its intermediate-only counterpart by 3 points, underscoring the critical role of specialization at the output level. Remarkably, with fine-grained design applied across both two dimensions, FineRMoE achieves the best average performance. It also exhibits superior parameter efficiency, attaining better performance with fewer activated parameters. Furthermore, as detailed in the analysis on Fig. 4 in Appendix D, the lowest average similarity among experts in each layer is achieved by FineRMoE, demonstrating the high specialization of its experts. These experiments systematically validate the effectiveness of the finer-grained design in our FineRMoE architecture, contributing to enhanced expert specialization and improved performance.

## 4.3 ABLATION STUDY ON FINERMOE ARCHITECTURE

To further explore the design criteria of the FineRMoE architecture, we compare two variants:

**AddConcatProj**: In the multi-head attention (Vaswani et al., 2017), a linear layer is adopted for projection after concatenating the outputs from each head. To explore its impact, we implement the variant by adding a linear projection layer after the concatenation operation in the FineRMoE.

**NoShareExpert**: In the above experiments, the pre-trained FFN is initialized as the shared expert. To explore its necessity, we implement the variant by removing the shared expert in the FineRMoE.

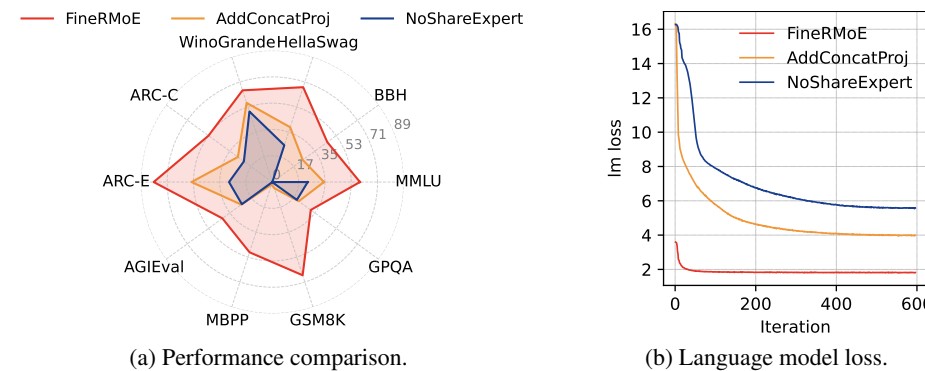

(a) Performance comparison.  (b) Language model loss.

Figure 2: The ablation study on the architecture of FineRMoE based on Qwen2.5-1.5B.

Table 3: The ablation study on fine-grained configurations based on Qwen2.5-1.5B.

| $G_I$ | $G_O$ | #Experts | #A-Experts | #Param/B | #A-Param/B | Inter-Dim | Out-Dim | Performance |
|---|---|---|---|---|---|---|---|---|
| **Pre-trained** | | - | - | 1.54 | 1.54 | 8960 | 1536 | 55.39 |
| 2 | 2 | 8 | 2 | 5.63 | 2.74 | 4480 | 768 | 53.71 |
| 4 | 2 | 16 | 2 | 5.63 | 2.26 | 2240 | 768 | 54.13 |
| 4 | 4 | 32 | 4 | 8.72 | 2.65 | 2240 | 384 | 55.33 |
| 8 | 2 | 32 | 2 | 5.63 | 2.02 | 1120 | 768 | 55.00 |
| 8 | 4 | 64 | 4 | 8.72 | 2.22 | 1120 | 384 | 56.09 |
| 16 | 2 | 64 | 2 | 5.64 | 1.90 | 560 | 768 | 55.45 |
| 16 | 4 | 128 | 4 | 8.72 | 2.00 | 560 | 384 | 55.98 |
| 16 | 8 | 256 | 8 | 14.90 | 2.21 | 560 | 192 | 56.08 |
| **32** | **2** | **128** | **2** | **5.64** | **1.85** | **280** | **768** | **56.21** |
| 32 | 4 | 256 | 4 | 8.74 | 1.91 | 280 | 384 | 56.25 |
| 32 | 8 | 512 | 8 | 14.92 | 2.03 | 280 | 192 | 56.01 |
| 64 | 2 | 256 | 2 | 5.65 | 1.83 | 140 | 768 | 56.14 |
| 64 | 4 | 512 | 4 | 8.76 | 1.88 | 140 | 384 | 56.26 |
| 64 | 8 | 1024 | 8 | 14.97 | 1.97 | 140 | 192 | 56.34 |

For an efficient study, the experiments of the variants are conducted based on Qwen2.5-1.5B by training on 10B tokens. From the performance comparison in Fig. 2 (a), FineRMoE consistently surpasses its two architectural variants by large margins on all evaluation benchmarks. As shown in Fig. 2 (b), at the beginning of training, FineRMoE exhibits a much lower initial language model (lm) loss compared to its two variants. Throughout the training process, FineRMoE demonstrates the quickest convergence speed, ultimately achieving the lowest lm loss value at the end of training.

The ablation results clearly demonstrate the effectiveness of our architectural design. First, adding a projection layer after concatenation harms performance, which may be due to the lack of effective initialization, thus leading to inadequate training of the linear layer. Second, the shared expert is essential, removing it causes poor convergence during training when using fine-grained sparse experts, which is consistent with the results achieved by S16A4 in Table 1.

## 4.4 ABLATION STUDY ON FINE-GRAINED CONFIGURATIONS

With fine-grained design applied across both intermediate and output dimensions, in this section, we delve into the impact of different fine-grained configurations on the performance. Based on the fixed setting of $R_I = 1, R_O = 2$, experiments are conducted by setting $G_I \in [2, 4, 8, 16, 32, 64]$ and $G_O \in [2, 4, 8]$ based on Qwen2.5-1.5B by training on 10B tokens for a quick study. The statistics of number of experts, the amount of parameters, and the average performance are presented in Table 3. Evaluation details are delivered in Appendix E.

Regarding the settings of $G_I$, the results show that the finer-grained MoEs with the intermediate granularity no fewer than 8 can exceed the performance of the pre-trained model through upcycling training. This underscores the importance of fine-grained design along the intermediate dimension in reducing expert redundancy and enhancing model performance. More notably, for each $G_I$ configuration, increasing $G_O$ from 2 to 8 contributes to a stable and consistent performance gain for

Table 4: The ablation study on router design based on Qwen2.5-1.5B.

| Settings | MMLU | BBH | Hell. | Wino. | ARCC. | ARCE. | AGIE. | MBPP | GSM8K | GPQA | AVG |
|---|---|---|---|---|---|---|---|---|---|---|---|
| **Separate Router** | 59.31 | 43.62 | 67.73 | 64.88 | 52.05 | 79.80 | 38.62 | 45.00 | 66.19 | 32.37 | 54.96 |
| **Single Router** | 59.30 | 45.97 | 67.51 | 65.27 | 53.41 | 80.35 | 41.81 | 50.00 | 66.34 | 32.14 | **56.21** |

the built MoE models. This further highlights that a higher output granularity is more conducive for promoting expert specialization and, thereby, strengthening overall performance.

Besides, ablation studies on the number of activated experts $T_I$, intermediate expansion rate $R_I$, and output expansion rate $R_O$ are analyzed in Appendix F. Considering the trade-off between parameter efficiency and model performance, we ultimately select the configuration of $G_I = 32$, $R_I = 1$, $G_O = 2$, $R_O = 2$ in the main experiments above.

### 4.5 ABLATION STUDY ON ROUTER DESIGN

To investigate the impact of the proposed router mechanism, which employs a single router to control the activation in the two sparse layers, we implement a version of FineRMoE with two separate routers. We train this version with 10B tokens for a quick validation. Results are presented in Table 4. Under the same data setting, employing the single-router design achieves better performance.

We infer the reason as that the single-router mechanism ensures that the vectors selected in the sparse concatenation layer are composed of outputs from experts with relatively higher routing scores. In contrast, using two separate routers could lead to a discrepancy: an expert highly scored by the router of sparse sum layer might not be selected by the router of the sparse concatenation layer. Consequently, the MoE layer's output might be dominated by experts with lower scores from the sparse sum layer, leading to performance degradation. This comprehensively validates the effectiveness of our unified router mechanism.

## 5 CONCLUSIONS

We propose the FineRMoE architecture, which pioneers the expansion of the fine-grained expert design in MoE models from only the intermediate dimension to the output dimension. The resulting finer-grained architecture consists of a sparse concatenation layer and a sparse sum layer, and employs a specially designed router mechanism to enable a single routing network to guide the activation of both two sparse layers. In addition, to facilitate efficient MoE model training, we propose an upcycling method not only applicable to the FineRMoE architecture, but also compatible with existing methods for constructing experts by duplicating or splitting pre-trained FFNs. Based on Qwen2.5 with sizes of 0.5B, 1.5B and 7B, we build the FineRMoE with 128 total experts and 2 activated via our upcycling method. Experiments demonstrate the proposed FineRMoE achieves the best performance on 10 benchmarks compared with baseline methods, as well as significant efficiency on both parameters and inference. Furthermore, extensive ablation studies validate the effectiveness of the finer-grained design in mitigating redundancy among experts for enhanced specialization and the proposed upcycling method for efficient MoE training.

**Limitations and Future Work.** By means of the proposed upcycling approach for efficient training, our FineRMoE architecture has proven effective in generalizing fine-grained expert design from intermediate to output dimensions. Yet, due to constraints in compute and data resources, the potential of FineRMoE in pre-training from scratch remains unexplored. Ongoing research efforts will address this gap to further corroborate the advantages of the FineRMoE architecture.

## ETHICS STATEMENT

This work presents the FineRMoE architecture and an associated upcycling method for efficient training. As a fundamental study in model design, it adheres to the ICLR Code of Ethics. The research involves no human subjects or sensitive data and raises no direct deployment concerns. The authors declare no conflicts of interest.

## REPRODUCIBILITY STATEMENT

To facilitate the reproducibility of the proposed FineRMoE and upcyling method, we have provided comprehensive details throughout this paper. The complete architecture and the novel router mechanism are formally defined in Sec. 3.1 and Sec. 3.2, respectively. The proposed upcycling method for efficient expert construction is detailed in Sec. 3.3. Our experimental setup, including model configurations, hyper-parameters, and training details, is thoroughly described in Sec. 4 and Appendix B. Moreover, to enable direct replication and further extension of our research, we commit to releasing our full source code implementation based on Megatron-LM framework and the pre-trained model weights. These resources will be made publicly available to the research community.

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

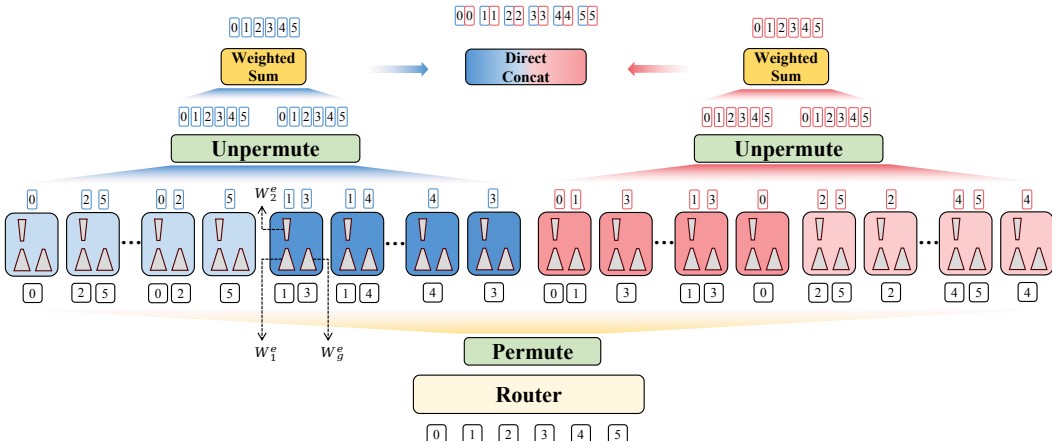

Figure 3: The forward computation process of a sequence of tokens in the sparse experts of Fin-eRMoE. For a given input sequence, the router first calculates the set of activated experts for each token. The tokens are then permuted to allow for parallel expert computation. After processing by the experts, the outputs are unpermuted to their original token order. For each token, the outputs from its activated experts are combined via a weighted sum to form dimension-reduced components. Finally, these components are concatenated to produce the final, dimension-restored output.

Table 5: The training hyper-parameters in experiments.

| Hyper-parameter | Setting |
|---|---|
| Optimizer | Adam |
| Adam_beta1 | 0.9 |
| Adam_beta2 | 0.95 |
| Clip Grad | 1.0 |
| Learning Rate | 1e-5 |
| Minimum Learning Rate | 1e-7 |
| LR Decay Style | Cosine |
| Precision | BF16 |
| Micro Batch Size | 1 |
| Global Batch Size | 2048 |
| Sequence Length | 8192 |
| Max Padding Length | 8192 |

## LLM USAGE STATEMENT

The writing of this paper is polished with the assistance of LLMs for grammar checking and text re-finement. Except for that, all core intellectual contributions, including the ideas, code development, experimental design, and result analysis, are conducted exclusively by the authors.

## A    FORWARD COMPUTATION PROCESS

Based on the router mechanism design in Sec. 3.2, the computation process of a sequence of tokens in the sparse experts of FineRMoE is demonstrated in Fig. 3.

## B    EXPERIMENTAL SETUP

### B.1    LOAD BALANCING LOSS

During the training phase, the FineRMoE is optimized with the sum of language modeling loss and the weighted load balancing loss. Specifically, we follow DeepSeek-V2 (Liu et al., 2024a) for the load balancing loss design. Given the number of experts as $N$, the number of activated experts per

token as $G_OT_I$, the number of tokens in a sequence as $L$, the score of assigning the $t$-th token to the $i$-th expert as $s_{i,t}$, the load balancing loss $L_{lbl}$ is calculated as:

$$
\begin{aligned}
L_{lbl} &= \alpha \sum_{i=1}^{N} f_i P_i, \\
P_i &= \frac{1}{L} \sum_{t=1}^{L} s_{i,t}, \\
f_i &= \frac{N}{G_O T_I L} \sum_{t=1}^{L} \mathbf{I}_{i,t}, \\
\mathbf{I}_{i,t} &= \begin{cases} 1, \ if \ Token \ t \ selects \ Expert \ i, \\ 0, \ else. \end{cases}
\end{aligned}
\tag{11}
$$

Throughout all experiments, the weight $\alpha$ of the load balancing loss is set as 0.001.

### B.2 IMPLEMENTATION DETAILS

We implement and train FineRMoE using the Megatron-LM [1] (Shoeybi et al., 2019) framework for its parallelization flexibility, based on the Qwen2.5 (Yang et al., 2024). The training data in all experiments is prepared by mixing and refining publicly available pre-training corpora, including English webpage data, Chinese webpage data, English knowledge data, Chinese knowledge data, code data, math data, the pile, wiki, book data. We set the data mixture ratio for each type equally. Aside from the different fine-grained and parallelization configurations, the data and detailed training settings are kept consistent across all experiments. For each training experiment, the number of warmup steps is 1% of the total training steps. Training hyper-parameters are summarized in Table 5.

We evaluate the models covering areas such as knowledge, reasoning, code, and math using the widely-recognized Language Model Evaluation Harness (Gao et al., 2024) framework. The evaluation benchmarks include: MMLU (Hendrycks et al., 2021), BBH (Suzgun et al., 2022), HellaSwag (Zellers et al., 2019), WinoGrande (Sakaguchi et al., 2021), ARC-Challenge (ARC-C) (Clark et al., 2018), ARC-Easy (ARC-E) (Clark et al., 2018), AGIEval (Zhong et al., 2023), MBPP (Austin et al., 2021), GSM8K (Cobbe et al., 2021), GPQA (Rein et al., 2024).

## C INFERENCE EFFICIENCY ANALYSIS

We complement the primary performance evaluation with an analysis of inference efficiency. Consistent with settings in Sec. 4.1, we select the C32A2, S16A4, Drop-Upcyling, NVShard and our FineRMoE, which are realized by continued training on 50B tokens based on Qwen2.5-7B, for comparison. Typically, model inference is divided into the prefill and decoding stages. For the prefill stage, the efficiency is evaluated by latency, specifically the Time to First Token (TTFT), which measures the duration from receiving the input to generating the first output token. A lower TTFT indicates higher prefill efficiency. For the decoding stage, the efficiency is evaluated by throughput, measured in tokens generated per second (tokens/s). A higher throughput indicates greater efficiency in the decoding stage.

Given the input "*Give me a short introduction to large language model.*", the comparisons are presented in Table 6. FineRMoE achieves the highest throughput. Owing to a parameter size far exceeding those of other methods, C32A2 demonstrates considerably inferior efficiency during both prefill and decoding stages, even though it achieves competitive performance to FineRMoE in Table 1. It is particularly noteworthy that FineRMoE outperforms C32A2 in benchmark performance while also delivering dramatically superior inference efficiency: a TTFT of 178.3 ms, which is 281 times faster than C32A2's 50245.9 ms, and a throughput of 27.3 tokens/s, which is 136 times higher than C32A2's 0.2 tokens/s. Although the smaller activation model size of S16A4 and NVShard affords them greater prefill efficiency compared to FineRMoE, they exhibit lower efficiency during

---

[1] https://github.com/NVIDIA/Megatron-LM/tree/6ba97dd37150a6bfba03d31808674211cf2a4d0d

Table 6: The comparisons of the inference efficiency between C32A2, S16A4, DU, NVShard and our FineRMoE. The models are constructed by continued training on 50B tokens on Qwen2.5-7B.

| Models | C32A2 | S16A4 | DU | NVShard | FineRMoE |
|---|---|---|---|---|---|
| **#Param (B)** | 184.42 | 7.62 | 47.54 | 47.55 | 26.65 |
| **#A-Param (B)** | 13.33 | 3.34 | 13.32 | 7.63 | 7.94 |
| **#Generated Tokens** | 107 | 47 | 32000 | 332 | 110 |
| **TTFT (ms) ↓** | 50245.9 | **78.5** | 84.8 | 137.8 | 178.3 |
| **Throughput (tokens/s) ↑** | 0.2 | 24.0 | 25.6 | 18.9 | **27.3** |
| ***Generated Text*** | | | | | |
| C32A2 | *Large language models (LLMs) are artificial intelligence (AI) models that are trained on vast amounts of text data to understand and generate human language. They are designed to recognize patterns in language, understand context, and generate responses that are coherent and contextually appropriate. LLMs are used in a wide range of applications, including natural language processing (NLP), machine translation, and text summarization. They are also used in conversational agents, such as chatbots and virtual assistants, to provide human-like interactions with users.* | | | | |
| S16A4 | *Large language models are a type of artificial intelligence that can generate text based on a vast amount of training data. They are designed to understand and generate human-like text, often used in chatbots, customer service, and other applications.* | | | | |
| DU | *Large language model (LLM) is a powerful tool for natural language processing, machine learning, and data science. It is a type of machine learning that uses a deep neural network to learn from data and make predictions. The goal of LLM is to understand and interpret human language, and to extract insights from it. Here is a short introduction to LLM:Introduction:LLM is a type of machine learning that uses a deep neural network to learn from data and make predictions. It is a popular tool for natural language processing, machine learning, and data science. LLM is a type of machine learning that uses a deep neural network to learn from data and make predictions. The goal of LLM is to understand and interpret human language, and to extract insights from it. [Repeat...]* | | | | |
| NVShard | *A large language model (LLM) is a type of artificial intelligence (AI) model that is designed to understand and generate human-like text. These models are typically based on deep learning techniques, particularly transformer architectures, which are capable of processing and understanding the context of large amounts of text data. LLMs are trained on vast datasets that include books, articles, websites, and other forms of text. During training, the model learns to predict the next word in a sequence based on the preceding words, which allows it to generate coherent and contextually relevant text. The size of the model, often measured in terms of the number of parameters it contains, is a key factor in its performance. Larger models tend to have more parameters and can capture more complex patterns in the data, leading to better performance on various natural language processing tasks.*
*Some of the most well-known large language models include GPT-3 (Generative Pre-trained Transformer 3) by OpenAI, BERT (Bidirectional Encoder Representations from Transformers) by Google, and T5 (Text-to-Text Transfer Transformer) by Google. These models have been used for a wide range of applications, including language translation, question-answering, text summarization, and even creative writing.*
*The development of large language models has been a significant advancement in the field of AI, as it has enabled machines to understand and generate human language in a way that was previously thought to be beyond the capabilities of computers. However, these models also raise important ethical and privacy concerns, as they can be used to generate misleading or harmful content and may inadvertently perpetuate biases present in the training data.* | | | | |
| FineRMoE | *Large language models (LLMs) are artificial intelligence (AI) systems that are trained on vast amounts of text data to understand and generate human-like language. They are designed to comprehend context, answer questions, and generate coherent and relevant text based on the input they receive. LLMs are trained using deep learning techniques, such as transformers, which allow them to process and generate text in a way that is similar to human language processing. They are used in a wide range of applications, including natural language processing, machine translation, and chatbots.* | | | | |

the decoding stage, and markedly poorer performance in Table 1. Furthermore, due to the partial re-initialization of experts, the model built by Drop-Upcycling suffers from severe performance degradation. This results in highly repetitive output that continues until it reaches the preset maximum length of 32000 tokens.

As comprehensively evidenced by the detailed comparisons in Table 1 and Table 6, the proposed FineRMoE achieves optimal performance across ten benchmarks, as well as maintaining remarkable efficiency in both parameters and inference. This solidly demonstrates the superiority of the FineRMoE architecture compared to existing ones.

## D  EXPERT SIMILARITY ANALYSIS

To provide an intuitive demonstration of the benefits conferred by the finer-grained design, we additionally evaluate the average similarity among the sparse experts in each layer. The analyzed

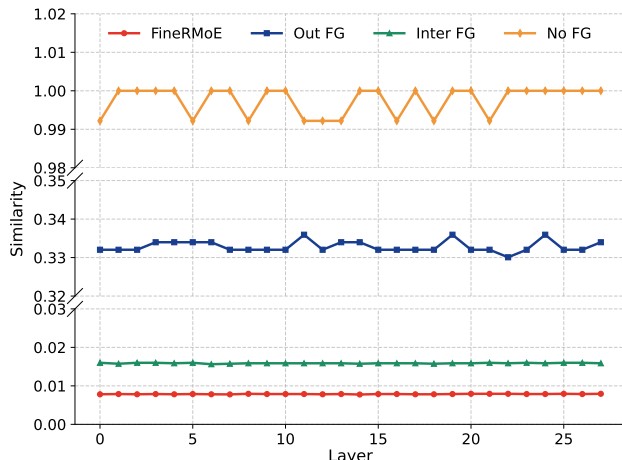

Figure 4: The average similarity among the sparse experts across all layers in the effectiveness validation of finer-grained design. The corresponding models are trained based on Qwen2.5-1.5B on 10B tokens for efficiency.

Table 7: Evaluation details across 10 benchmarks in the ablation study on fine-grained configurations based on Qwen2.5-1.5B.

| $G_I$ | $G_O$ | MMLU | BBH | HellaSwag | WinoGrande | ARC-C | ARC-E | AGIEval | MBPP | GSM8K | GPQA |
|---|---|---|---|---|---|---|---|---|---|---|---|
| 2 | 2 | 58.19 | 42.45 | 66.56 | 64.09 | 51.19 | 78.28 | 40.41 | 46.00 | 63.84 | 26.12 |
| 4 | 2 | 58.15 | 43.46 | 66.44 | 63.69 | 51.88 | 80.13 | 41.80 | 44.40 | 63.46 | 27.90 |
| 4 | 4 | 58.97 | 44.68 | 67.14 | 65.11 | 51.62 | 80.30 | 41.15 | 47.20 | 66.11 | 31.03 |
| 8 | 2 | 58.16 | 45.48 | 66.79 | 64.48 | 50.77 | 79.12 | 39.85 | 46.60 | 65.50 | 33.26 |
| 8 | 4 | 59.17 | 47.00 | 67.46 | 64.88 | 53.24 | 80.39 | 42.16 | 50.00 | 66.03 | 30.58 |
| 16 | 2 | 59.06 | 45.51 | 67.24 | 64.01 | 52.82 | 79.88 | 41.55 | 48.00 | 66.11 | 30.36 |
| 16 | 4 | 59.56 | 46.14 | 67.72 | 64.01 | 54.01 | 80.47 | 42.37 | 47.00 | 66.41 | 32.14 |
| 16 | 8 | 59.53 | 45.72 | 67.52 | 64.64 | 53.24 | 80.51 | 42.14 | 50.40 | 66.03 | 31.03 |
| 32 | 2 | 59.30 | 45.97 | 67.51 | 65.27 | 53.41 | 80.35 | 41.81 | 50.00 | 66.34 | 32.14 |
| 32 | 4 | 59.58 | 46.29 | 67.56 | 65.67 | 53.58 | 80.30 | 42.48 | 50.20 | 67.63 | 29.24 |
| 32 | 8 | 59.05 | 46.24 | 67.67 | 64.64 | 52.90 | 80.35 | 41.56 | 48.80 | 66.72 | 32.14 |
| 64 | 2 | 59.31 | 45.57 | 67.81 | 64.80 | 53.33 | 80.43 | 42.51 | 48.80 | 67.55 | 31.25 |
| 64 | 4 | 59.26 | 45.65 | 67.75 | 64.40 | 52.73 | 80.43 | 41.64 | 50.20 | 67.02 | 33.48 |
| 64 | 8 | 59.75 | 46.43 | 67.81 | 65.43 | 52.56 | 80.18 | 41.41 | 50.80 | 66.03 | 33.04 |

MoE models are trained via the four distinct settings in Sec. 4.2, i.e., fine-grained design only on intermediate dimension, output dimension, both the two dimensions, and no fine-grained design. Specifically, we enumerate all pairwise combinations of experts in each layer, calculate their cosine similarities, and then average across all combinations to derive the average expert similarity for all layers under each configuration, results are delivered in Fig 4.

Notably, with the lack of fine-grained design, the model obtained by simply replicating the pre-trained FFNs demonstrates the highest expert similarity, indicating severe redundancy and explaining its poorest performance in Table 2. When fine-grained design is applied to only one dimension, the expert similarity is lower. Given that the intermediate granularity $G_I = 32$ is greater than the output granularity $G_O = 2$, applying the fine-grained design only to the output dimension leads to a slightly higher degree of expert similarity than applying it to the intermediate dimension. It is noteworthy that our proposed FineRMoE architecture, which employs the fine-grained design on both the intermediate and output dimensions, exhibits the lowest level of expert similarity across all four settings. This signifies that the experts have attained alleviated redundancy and a high degree of specialization, thereby leading to the superior performance in Table 2.

# E  DETAILED EVALUATION ON FINE-GRAINED CONFIGURATIONS

The detailed evaluation results across ten benchmarks in the ablation study of fine-grained configurations are delivered in Table 7.

Table 8: The ablation study on $T_I$, $R_I$, $R_O$ based on Qwen2.5-1.5B by training on 10B tokens.

| Settings | Pre-Trained | Base | Ablation on $T_I$ | | Ablation on $R_I$ | | Ablation on $R_O$ | |
|---|---|---|---|---|---|---|---|---|
| $G_I,R_I,G_O,R_O$ | - | 32,1,2,2 | 32,1,2,2 | 32,1,2,2 | 32,2,2,2 | 32,4,2,2 | 32,1,2,4 | 32,1,2,8 |
| $T_I$ | - | 1 | 2 | 4 | 1 | 1 | 1 | 1 |
| #Experts | - | 128 | 128 | 128 | 256 | 512 | 256 | 512 |
| #A-Experts | - | 2 | 4 | 8 | 2 | 2 | 2 | 2 |
| #Param / B | 1.54 | 5.64 | 5.64 | 5.64 | 9.51 | 17.24 | 9.51 | 17.24 |
| #A-Param / B | 1.54 | 1.85 | 1.91 | 2.03 | 1.86 | 1.88 | 1.86 | 1.88 |
| MMLU | 60.87 | 59.30 | 59.37 | 59.76 | 59.14 | 59.68 | 59.44 | 58.99 |
| BBH | 43.33 | 45.97 | 46.97 | 46.11 | 45.38 | 45.12 | 46.15 | 45.72 |
| HellaSwag | 67.84 | 67.51 | 67.80 | 67.81 | 67.42 | 67.42 | 67.35 | 67.26 |
| WinoGrande | 64.88 | 65.27 | 63.85 | 64.33 | 64.72 | 65.59 | 63.93 | 65.11 |
| ARC-C | 54.86 | 53.41 | 52.90 | 53.41 | 53.92 | 53.07 | 53.33 | 51.88 |
| ARC-E | 81.02 | 80.35 | 80.47 | 80.35 | 79.97 | 80.05 | 80.35 | 79.84 |
| AGIEval | 39.83 | 41.81 | 41.87 | 42.03 | 41.68 | 41.45 | 41.64 | 41.84 |
| MBPP | 43.40 | 50.00 | 49.20 | 49.40 | 51.40 | 48.40 | 49.20 | 46.60 |
| GSM8K | 65.73 | 66.34 | 65.73 | 66.34 | 67.17 | 66.94 | 67.10 | 66.94 |
| GPQA | 32.14 | 32.14 | 32.59 | 33.71 | 31.92 | 33.26 | 31.03 | 33.04 |
| Average | 55.39 | 56.21 | 56.08 | 56.33 | 56.27 | 56.10 | 55.95 | 55.72 |

# F  ABLATION STUDY ON $T_I$, $R_I$, $R_O$

In our main experiments, we aim to achieve superior performance with high parameter efficiency for the FineRMoE model. To this end, while maintaining sparsity in both the concatenation and sum layers, we adopt the following settings to minimize both the total and activated parameters: the number of activated experts per group in the sparse sum layer $T_I = 1$, the intermediate expansion rate $R_I = 1$, and the output expansion rate $R_O = 2$.

In this section, we further explore the impact of $T_I$, $R_I$, $R_O$ to the performance compared with the base configuration of $G_I = 32$, $R_I = 1$, $G_O = 2$, $R_O = 2$. Specifically, by setting $T_I \in [2, 4]$, $R_I \in [2, 4]$, $R_O \in [4, 8]$, the ablation experiments are performed based on Qwen2.5-1.5B with 10B tokens trained for a quick investigation. Results are summarized in Table 8.

Compared with the pre-trained model, performance achieved by all ablation settings consistently exhibits remarkable advantages. Experimental results reveal that under the base fine-grained configuration, performance improvements can be stably achieved despite adjustments to $T_I$, $R_I$, $R_O$, thereby further corroborates the effectiveness of the proposed FineRMoE architecture and the upcycling method for efficient training. Compared with the base configuration, altering $T_I$, $R_I$, $R_O$ leads to minor performance fluctuations. Therefore, considering the trade-off between performance gains and parameter efficiency, employing the base configuration represents the optimal choice.

