# OpenReview forum: "FineRMOE: Dimension Expansion for Finer-Grained Expert with Its Upcycling Approach"
_ICLR.cc/2026/Conference — Submitted to ICLR 2026_

### Official Review · Reviewer_WeGh · 2025-10-21

**Soundness:** 3
**Presentation:** 3
**Contribution:** 2
**Rating:** 4
**Confidence:** 5

**Summary:**

This paper proposes FineRMoE, a Mixture-of-Experts (MoE) model that introduces fine-grained sparsity not only in the intermediate dimension but also in the output dimension.
The method aims to improve both parameter and training efficiency by refining sparsity across multiple levels.

In upcycling from a dense model to an MoE model, FineRMoE shows less than one-point improvement on average across 10 benchmark tasks, and the authors claim that the model shows good performance compared to existing upcycling methods.

**Strengths:**

- The paper is clearly written overall, and the figures and pseudocode make the method easy to follow.

- The method implemented within Megatron-LM, which is a strong practical advantage.

**Weaknesses:**

1. The results are incremental, the improvement is modest (less than one point on average over 10 benchmarks).

2. The paper lacks sufficient details for reproducibility, such as dataset specification, hyperparameters (e.g., load balancing loss, warmup steps), and environment details (e.g., Megatron-LM version).

3. There is a minor typo in Algorithm 1 (line 219): ${G_I}$: intermediate expansion rate should be ${R_I}$: intermediate expansion rate.

4. Since the main claim is efficiency for upcycling under limited compute, the authors should report actual GPU hours and FLOPs to verify the computational advantage.

5. Although FineRMoE aims to improve efficiency, the proposed router mechanism introduces additional overhead, which may offset the theoretical gains in real training scenarios.

6. Table 3 indicates that the model’s performance strongly depends on the hyperparameters (especially granularity settings).

7. The overall improvement may not justify the added complexity, the cost-effectiveness of the method remains unclear.

8. A comparison with prior fine-grained upcycling work (e.g., arXiv:2410.07524) would be informative.

**Questions:**

1. Benchmark selection: Why did you choose the particular benchmarks used for evaluation?
The Qwen 2.5 paper (arXiv:2412.15115) includes MATH, HumanEval, and MMLU-Pro. Since results vary across tasks, could benchmark choice affect the perceived improvement?

2. Why not Qwen 3? Experiments are limited to Qwen 2.5. Is there a reason for not testing on Qwen 3? Would the proposed method remain effective on stronger or more modern base models?

3. Scratch training feasibility: The paper claims that FineRMoE can also be trained from scratch. Could you demonstrate this on a smaller setup (e.g., fewer tokens or smaller model) to show practicality for users? This would be valuable since many current models (e.g., Qwen 3, DeepSeek V3) are trained from scratch rather than upcycled.

4. Router design: Would using two separate routers (one for the intermediate and one for the output) yield better performance than the shared single-router setup? Have you investigated this variant?

---

> ### Author Response · Authors · 2025-11-20
> **Response Part 1.**
>
> We appreciate your comments and valuable advice on improving our manuscript. We hope that the responses provided below will address all your concerns.
>
> - **Response of W1:**
> 1. **Compared to PT, CT, and S16A4**, FineRMoE achieves substantial performance improvements across 10 benchmarks covering general, mathematical, and coding tasks. Specifically, FineRMoE outperforms PT and CT over 1 point on sizes of 1.5B and 7B. Moreover, FineRMoE attains an average **performance improvement of at least 70\%** over S16A4 across all three model sizes.
> 2. **Compared to C32A2**, FineRMoE not only outperforms it across 10 benchmarks, but also achieves this with **superior parameter and inference efficiency**. Specifically, it **owns fewer than $\frac{1}{6}$ of the total parameters of C32A2** across three model sizes. Furthermore, as shown in Table 6, FineRMoE's **TTFT** (178.3 ms) is **281 times faster** than that of C32A2 (50245.9 ms), and FineRMoE's **throughput** (27.3 tokens/s) is **136 times higher** than that of C32A2 (0.2 tokens/s).
> 3. In summary, FineRMoE demonstrates significant superiority over baseline methods on **benchmark performance, parameter efficiency, and inference efficiency**.
>
> - **Response of W2:**
>
> Thank you for your valuable suggestion.
> 1. As for **dataset specification**, our training dataset includes English webpage data, Chinese webpage data, English knowledge data, Chinese knowledge data, code data, math data, the pile, wiki, book data. We set the data mixture ratio for each type equally.
> 2. As for the **load balancing loss**, we follow DeepSeek-V2 for the load balancing loss design. Given the number of experts as $N$, the number of activated experts per token as $G_OT_I$, the number of tokens in a sequence as $L$, the score of assigning the $t$-th token to the $i$-th expert as $s_{i,t}$, the load balancing loss $L_{lbl}$ is calculated as:
> $$\begin{aligned}
> L_{lbl} = \alpha\sum_{i=1}^{N}f_iP_i, \\
> P_i =\frac{1}{L}\sum_{t=1}^{L}s_{i,t}, \\
> f_i = \frac{N}{G_OT_IL} \sum_{t=1}^{L}\mathbf{I}_{i,t},
> \end{aligned}$$
>
> where $\mathbf{I}_{i,t}$ is an indicator function, which equals to 1 when the $t$-th token selects the $i$-th expert, otherwise 0. Specifically, the FineRMoE is optimized with the sum of language modeling loss and the weighted load balancing loss, and the weight of load balancing loss $\alpha$ is set as 0.001 throughout our experiments.
>
> 3. As for the **warmup steps**, for each training experiment, the number of warmup steps is 1\% of the total training steps.
>
> 4. As for the **Megatron-LM**, throughout all experiments, the implementation and training are based on the version as: https://github.com/NVIDIA/Megatron-LM/tree/6ba97dd37150a6bfba03d31808674211cf2a4d0d.
>
> 5. We have added the above demonstration in Appendix B in the updated PDF.
>
> - **Response of W3:**
>
> Thank you for your careful review, we have revised it in the updated PDF.
>
> - **Response of W4:**
> 1. Based on Qwen2.5-1.5B, the training time of the MoE models are: C32A2 with 1376 GPU hours, S16A4 with 880 GPU hours, and FineRMoE with 1126 GPU hours.
> 2. In addition, to verify the efficiency for upcycling under limited compute, we conduct a set of experiments to compare the performance achieved by the baselines and FineRMoE under equal computational budget. Specifically, we perform the experiments based on Qwen2.5-1.5B using **4 nodes (32 GPUs) and training for 16 hours** with the Megatron-LM framework. For C32A2, S16A4 and FineRMoE, the parallelism are kept the same: TP=1, PP=2, CP=1, ETP=1, EP=8. For CT, the parallelism setting is TP=1, PP=1. The results are presented below. According to the results, under equal limited computational cost, FineRMoE outperforms the baselines and achieves the best average performance. It further demonstrates the computational advantage of the proposed FineRMoE in continual training.
>
> | 32 GPUs, 16 Hours | PT    | CT    | C32A2 | S16A4 | FineRMoE |
> | ----------------- | ----- | ----- | ----- | ----- | -------- |
> | MMLU              | 60.87 | 60.63 | 61.01 | 25.94 | 59.61    |
> | BBH               | 43.33 | 45.09 | 45.40 | 21.10 | 46.34    |
> | HellaSwag         | 67.84 | 68.55 | 68.82 | 33.43 | 68.11    |
> | WinoGrande        | 64.88 | 65.19 | 65.59 | 52.25 | 65.11    |
> | ARC-C             | 54.86 | 53.67 | 53.58 | 24.15 | 52.99    |
> | ARC-E             | 81.02 | 80.77 | 80.81 | 46.21 | 80.51    |
> | AGIEval           | 39.83 | 38.91 | 38.85 | 25.75 | 40.83    |
> | MBPP              | 43.40 | 42.20 | 45.00 | 0.00  | 48.20    |
> | GSM8K             | 65.73 | 66.34 | 66.79 | 1.36  | 67.25    |
> | GPQA              | 32.14 | 31.70 | 30.58 | 22.10 | 32.14    |
> | Average           | 55.39 | 55.31 | 55.64 | 25.23 | 56.11    |

---

> > ### Author Response · Authors · 2025-11-20
> > **Response Part 2.**
> >
> > - **Response of W5:**
> >
> > As a supplementary experiment, we implement a version of FineRMoE with two separate routers. We compare the training time of our proposed single-router design with that of using separate routers for the two sparse layers. Based on Qwen2.5-1.5B with 10B tokens trained, the single-router approach requires 279 GPU hours, compared to 285 GPU hours cost by the two-router setup. The comparable training time demonstrates that our proposed router mechanism introduces ignorable additional overhead.
> >
> > - **Response of W6:**
> > 1. According to the analysis in Sec. 4.4 (Table 3) and Appendix F (Table 8), except for the first four fine-grained settings in Table 3, all other settings in that table and all configurations in Table 8 achieve better performance than the pre-trained model. Therefore, our method requires no specific hyperparameter tuning and can be readily deployed to improve performance.
> > 2. Moreover, this work aims to achieve a finer-grained expert design compared to existing MoE architectures. According to the analysis in Sec. 4.4 (Table 3), larger settings of $G_I$ and $G_O$ generally lead to stronger model performance and relatively fewer total/activated parameters. Therefore, in practice, the fine-grained hyperparameters can be set to relatively higher values to enhance parameter efficiency and overall performance.
> >
> > - **Response of W7:**
> > 1. As for the **training phase**, as validated in the **Response of W4**, FineRMoE achieves the best performance among all baselines under an equal, limited computational budget.
> > 2. As for the **inference phase**, as shown in Table 6 and demonstrated in the **Response of W1**, FineRMoE achieves the highest decoding throughput, and the latency of FineRMoE is even hundreds of times faster than C32A2.
> > 3. In summary, FineRMoE exhibits verifiable cost-effectiveness in both training and inference phases.
> >
> > - **Response of W8:**
> > 1. Thank you for your suggestion, based on our generalizable upcyling method, we are readily to implement the method(2410.07521), referred as NVShard. We conduct experiments based on Qwen2.5 with sizes of 0.5B, 1.5B and 7B by training on 50B tokens in consistency with the experiments in Table 1. Results are presented below.
> >
> > | size        | 0.5B  | 1.5B  | 7B    |
> > | ----------- | ----- | ----- | ----- |
> > | \#Experts   | 64    | 64    | 64    |
> > | \#A-Experts | 8     | 8     | 8     |
> > | \#Param/B   | 2.83  | 9.88  | 47.55 |
> > | \#A-Param/B | 0.63  | 1.78  | 7.63  |
> > | MMLU        | 39.49 | 56.96 | 69.81 |
> > | BBH         | 27.98 | 41.13 | 66.67 |
> > | HellaSwag   | 49.44 | 66.72 | 79.05 |
> > | WinoGrande  | 55.56 | 63.30 | 74.27 |
> > | ARC-C       | 34.56 | 49.49 | 59.81 |
> > | ARC-E       | 65.40 | 77.82 | 84.97 |
> > | AGIEval     | 26.79 | 36.78 | 49.36 |
> > | MBPP        | 16.80 | 33.80 | 53.20 |
> > | GSM8K       | 26.84 | 64.67 | 83.62 |
> > | GPQA        | 28.57 | 31.03 | 31.03 |
> > | Average     | 37.14 | 52.14 | 65.18 |
> >
> > 2. **Benchmark Performance.** Compared with the results in Table 1, FineRMoE achieves an average performance advantage of around 5 points over NVShard across three model sizes, demonstrating the effectiveness of our method.
> > 3. **Parameter Efficiency.** Besides, despite comparable activated parameter counts, NVShard contains about twice the total parameters of FineRMoE, highlighting the superior parameter efficiency of our method.
> > 4. Furthermore, we implement the **Drop-Upcycling** based on Qwen2.5 for a comprehensive comparison. Please refer to **Response of W1 to Reviewer j9xD** for details. Results are also presented as below. Drop-Upcycling achieves the performance far inferior to that of FineRMoE. The amount of total and activated parameters of Drop-Upcycling are both larger than FineRMoE. The experiments further demonstrate the superior performance and parameter-efficiency of the proposed FineRMoE. We have added the results of NVShard and Drop-Upcycling in the Sec. 4.1 and Table 1 for a more comprehensive baseline comparison. Besides, the inference efficiency of NVShard and Drop-Upcycling are added to the Appendix C and Table 6 for comparisons.
> >
> > | size        | 0.5B  | 1.5B  | 7B    |
> > | ----------- | ----- | ----- | ----- |
> > | \#Experts   | 8     | 8     | 8     |
> > | \#A-Experts | 2     | 2     | 2     |
> > | \#Param/B   | 2.83  | 9.87  | 47.54 |
> > | \#A-Param/B | 0.94  | 2.93  | 13.32 |
> > | MMLU        | 24.36 | 25.93 | 25.71 |
> > | BBH         | 6.93  | 10.17 | 17.98 |
> > | HellaSwag   | 28.51 | 30.07 | 34.07 |
> > | WinoGrande  | 50.67 | 52.41 | 51.85 |
> > | ARC-C       | 21.93 | 20.99 | 25.51 |
> > | ARC-E       | 40.70 | 43.27 | 51.30 |
> > | AGIEval     | 25.86 | 25.52 | 25.65 |
> > | MBPP        | 0.00  | 0.00  | 0.00  |
> > | GSM8K       | 2.27  | 1.90  | 2.81  |
> > | GPQA        | 25.00 | 24.78 | 23.66 |
> > | Average     | 22.62 | 23.50 | 25.85 |

---

> > > ### Author Response · Authors · 2025-11-20
> > > **Response Part 3.**
> > >
> > > - **Response of Q1:**
> > > 1. The 10 benchmarks employed in this paper already constitute a set designed to assess model capabilities in general tasks, coding, and math.
> > > 2. As a supplementary, we provide additional evaluation for the set of models built on Qwen2.5-1.5B in Table 1. Results are presented as below. FineRMoE achieves better performance advantage compared with baselines on these benchmarks, further proving the effectiveness of FineRMoE.
> > >
> > > | Benchmarks    | PT    | CT    | C32A2 | S16A4 | FineRMoE |
> > > | ------------- | ----- | ----- | ----- | ----- | -------- |
> > > | MMLU-Pro      | 29.13 | 30.31 | 8.39  | 8.24  | 31.21    |
> > > | HumanEval     | 37.20 | 51.22 | 62.20 | 7.32  | 55.49    |
> > > | CMath         | 60.75 | 67.94 | 68.49 | 3.46  | 70.58    |
> > > | HendrycksMath | 13.50 | 12.92 | 12.52 | 1.82  | 12.42    |
> > > | Average       | 35.15 | 40.60 | 37.90 | 5.21  | 42.43    |
> > >
> > > - **Response of Q2:**
> > > 1. The baseline methods are built based on models such as T5 (C32A2), LLaMA2-7B (S16A4), LLaMA and Mixtral (Drop-Upcycling), and Nemotron (NVShard). In contrast, we adopt Qwen2.5, which currently represents the strongest base model in the field of model upcycling.
> > > 2. As a supplementary, we build FineRMoE based on Qwen3-0.6B with 10B tokens trained for a quick validation. Results are presented below. The average performance of FineRMoE has been improved compared with the pre-trained Qwen3-0.6B, proving that FineRMoE is also effective on stronger and more modern base models.
> > >
> > > | Qwen3-0.6B | MMLU  | BBH   | HellaSwag | WinoGrande | ARC-C | ARC-E | AGIEval | MBPP  | GSM8K | GPQA  | Average |
> > > | ---------- | ----- | ----- | --------- | ---------- | ----- | ----- | ------- | ----- | ----- | ----- | ------- |
> > > | PT         | 47.19 | 41.41 | 46.62     | 55.96      | 40.19 | 71.38 | 34.19   | 22.60 | 51.40 | 29.02 | 44.00   |
> > > | FineRMoE   | 47.61 | 42.13 | 49.61     | 58.48      | 40.36 | 70.79 | 32.20   | 26.00 | 48.45 | 29.69 | 44.53   |
> > >
> > > - **Response of Q3:**
> > >
> > > Thank you for your suggestion. We choose the model configuration as the FineRMoE built based on the Qwen2.5-0.5B for training from scratch with 100B tokens. For a quick validation, the experiment adopts the same data and training strategies as in Table 1, **without any ablation experiments on data mixture and training scheduler**. Besides, we select the official Pythia-1B, which has been trained on 300B tokens, for a comparison. Results are presented as below. Compared to Pythia, FineRMoE exhibits a mere 8-point average performance gap while **using only one-third of the training data**. This result indicates the significant potential of the FineRMoE architecture for training-from-scratch, which we will explore in depth in our future work.
> > >
> > > |                  | Model-Size   | Train Tokens | MMLU  | BBH   | HellaSwag | WinoGrande | ARC-C | ARC-E | AGIEval | MBPP | GSM8K | GPQA  | Average |
> > > | ---------------- | ------------ | ------------ | ----- | ----- | --------- | ---------- | ----- | ----- | ------- | ---- | ----- | ----- | ------- |
> > > | FineRMoE-scratch | 1.68B-A0.65B | 100B         | 27.30 | 0.00  | 26.00     | 50.51      | 23.46 | 30.60 | 25.85   | 0.00 | 1.14  | 25.67 | 21.05   |
> > > | Pythia-1B        | 1B           | 300B         | 26.12 | 25.83 | 47.72     | 52.25      | 30.38 | 60.14 | 25.23   | 2.60 | 2.88  | 23.88 | 29.70   |
> > >
> > > - **Response of Q4:**
> > > 1. Thank you for your suggestion. To investigate this variant, we implement a version of FineRMoE with two separate routers. We train this version with 10B tokens for a quick validation. Results are presented below. Under the same data setting, employing the single-router design achieves better performance.
> > > 2. We infer the reason as that the single-router mechanism ensures that the vectors selected in the sparse concatenation layer are composed of outputs from experts with relatively higher routing scores. In contrast, using two separate routers could lead to a discrepancy: an expert highly scored by the router of sparse sum layer might not be selected by the router of the sparse concatenation layer. Consequently, the MoE layer's output might be dominated by experts with lower scores from the sparse sum layer, leading to performance degradation. This comprehensively validates the effectiveness of our unified router design. We have added this experiment and analysis in the Sec. 4.5 in the updated PDF.
> > >
> > > | Settings        | MMLU  | BBH   | HellaSwag | WinoGrande | ARC-C | ARC-E | AGIEval | MBPP  | GSM8K | GPQA  | Average |
> > > | --------------- | ----- | ----- | --------- | ---------- | ----- | ----- | ------- | ----- | ----- | ----- | ------- |
> > > | Separate-Router | 59.31 | 43.62 | 67.73     | 64.88      | 52.05 | 79.80 | 38.62   | 45.00 | 66.19 | 32.37 | 54.96   |
> > > | Single-Router   | 59.30 | 45.97 | 67.51     | 65.27      | 53.41 | 80.35 | 41.81   | 50.00 | 66.34 | 32.14 | 56.21   |

---

> > > > ### Comment · Reviewer_WeGh · 2025-11-25
> > > >
> > > > Thank you for your detailed responses.
> > > >
> > > > - Regarding Response of W8
> > > >
> > > > For the NVShard results, could you clarify why its performance is lower than the dense baseline?
> > > >
> > > > - Regarding Response of Q3
> > > >
> > > > Since MMLU, ARC-C, ARC-E, HellaSwag, AGIEval, and GPQA are all four-choice tasks, and WinoGrande is a two-choice task, the reported scores are essentially at chance level. This suggests that the model has not learned in a meaningful way, and I am not fully convinced that the proposed method is effective.
> > > > Given the 100B-token budget, I would normally expect performance above chance on tasks such as HellaSwag.

---

> > > > > ### Author Response · Authors · 2025-11-27
> > > > > **Response to WeGh Part 1**
> > > > >
> > > > > Thank you for your positive reply and the engaging discussion, which will undoubtedly help us refine our work. In response to the two new questions you raised concerning NVShard and training from scratch, we have provided detailed responses below and hope they will fully address all your concerns.
> > > > >
> > > > > - **Response to the question of NVShard**
> > > > > 1. In contrast to the 1T tokens used for training in the NVShard paper, we implement it by training on 50B tokens in our work for a fair comparison with other methods.
> > > > > 2. We hold that the training process of NVShard comprises two phases: an initial alignment phase and an architectural adaptation phase. Specifically, in the **initial alignment phase**, the virtual group design ensures that the weighted fusion of activated expert outputs in NVShard closely approximates the output of the pre-trained dense FFN, thereby stabilizing the initial loss. In our experiment, when training on 50B tokens based on Qwen2.5-1.5B, NVShard achieved an initial LM loss of 2.65, compared to 3.61 for FineRMoE.
> > > > > 3. During the **architectural adaptation phase**, evolving parameters of the router and experts cause the MoE layer's output to deviate from the original FFN's. Consequently, this design necessitates more data to adequately train the routers and experts for optimal performance. With only 50B tokens, the model is likely still in the early stages of this adaptation. In our experiment, after training on 50B tokens with Qwen2.5-1.5B, NVShard's final LM loss is 1.85, whereas FineRMoE achieved a lower loss of 1.63.
> > > > > 4. Furthermore, the NVShard paper only reports evaluation results on the MMLU benchmark after training on 1T tokens. For experiments with smaller data volumes (110B tokens), only validation loss is provided. It is hard to know how it performs on additional benchmarks after training with fewer tokens. Combined with the previous analysis, we argue that NVShard requires larger-scale data to achieve performance improvements.

---

> > > > > ### Author Response · Authors · 2025-11-27
> > > > > **Response to WeGh Part 2**
> > > > >
> > > > > - **Response to the question of training from scratch**
> > > > > 1. Training from scratch necessitates extensive experiments on training strategies (e.g., learning rate, scheduler) and data mixtures. **The configurations for continual training may not be directly transferable.** Due to the limited computation, it is challenging for us to perform detailed ablation studies at this stage. Nevertheless, though Pythia-1B has been trained on three times of data compared with ours, FineRMoE performs comparably to or even surpasses it on MMLU, WinoGrande, ARC-C, AGIEval, and GPQA.
> > > > > 2. To deeply analyze whether the FineRMoE is effective in training from scratch, we conduct two sets of experiments. In the first experiment, we keep the training and data setting the same, and perform training from scratch using the Qwen2.5-0.5B architecture, which is known to be effective. In the second experiment, we increase the mixture ratio of English knowledge data, wiki, book to twice that of the initial experiment.
> > > > > 3. Under the same data mixture and training setting, the performance of FineRMoE and Qwen2.5-0.5B after training from scratch on 100B tokens is compared below. The results indicate that under identical data and training settings, Qwen2.5-0.5B, the architecture of which is known to be effective, underperforms FineRMoE. This suggests that the reported scores being at chance level may be attributable to the suboptimal data configuration or training strategies, rather than the ineffectiveness of FineRMoE. The marginal performance gain achieved by FineRMoE over the Qwen2.5-0.5B underscores our potential in training from scratch.
> > > > >
> > > > > | Train from Scratch on 100B tokens | MMLU  | BBH  | HellaSwag | WinoGrande | ARC-C | ARC-E | AGIEval | MBPP | GSM8K | GPQA  | Average |
> > > > > | --------------------------------- | ----- | ---- | --------- | ---------- | ----- | ----- | ------- | ---- | ----- | ----- | ------- |
> > > > > | Qwen2.5-0.5B                      | 25.22 | 0.32 | 25.59     | 48.54      | 23.12 | 28.54 | 25.33   | 0.00 | 2.20  | 22.77 | 20.16   |
> > > > > | FineRMoE-1.68B-A0.65B             | 27.30 | 0.00 | 26.00     | 50.51      | 23.46 | 30.60 | 25.85   | 0.00 | 1.14  | 25.67 | 21.05   |
> > > > >
> > > > > Furthermore, we compare the LM loss at the beginning and end of the training process as below. When trained from scratch on 100B tokens, FineRMoE and Qwen2.5-0.5B started with almost identical initial LM loss. However, under the same training and data settings, the training process of Qwen2.5-0.5B ended with a final LM loss of 4.26, which is significantly higher than FineRMoE's 3.76. This indicates that FineRMoE enables more effective training and achieves better convergence. In addition to the marginal performance advantage across the 10 benchmarks, the better convergence also demonstrates FineRMoE's potential for pre-training from scratch.
> > > > >
> > > > > | LM Loss               | Begin | End  |
> > > > > | --------------------- | ----- | ---- |
> > > > > | Qwen2.5-0.5B          | 11.95 | 4.26 |
> > > > > | FineRMoE | 11.96 | 3.76 |
> > > > >
> > > > > 4. FineRMoE's performance under the initial and new data settings is compared below. With an adjusted data mixture, a slight performance improvement is observed, demonstrating the critical role of data configuration in training from scratch.
> > > > >
> > > > > | Data settings | MMLU  | BBH  | HellaSwag | WinoGrande | ARC-C | ARC-E | AGIEval | MBPP | GSM8K | GPQA  | Average |
> > > > > | ------------ | ----- | ---- | --------- | ---------- | ----- | ----- | ------- | ---- | ----- | ----- | ------- |
> > > > > | Initial      | 27.30 | 0.00 | 26.00     | 50.51      | 23.46 | 30.60 | 25.85   | 0.00 | 1.14  | 25.67 | 21.05   |
> > > > > | New          | 28.83 | 0.49 | 30.12     | 50.28      | 27.13 | 30.51 | 27.82   | 0.00 | 2.21  | 29.29 | 22.67   |
> > > > >
> > > > > 5. In summary, our analysis indicates that FineRMoE has considerable potential for training from scratch. We acknowledge the current limitation of insufficient computational resources for a comprehensive investigation. Future work will address this through systematic experiments on data, training strategies, and scaling laws to train FineRMoE models from scratch.
> > > > >
> > > > >
> > > > > We thank you once again for the time and effort you have dedicated to reviewing our work. Your valuable comments are highly significant for helping us improve this work. Should you have any further questions, please do not hesitate to let us know. We eagerly await your response.

---

> > > > > > ### Comment · Reviewer_WeGh · 2025-11-27
> > > > > >
> > > > > > Thank you for the detailed responses and additional experiments.
> > > > > > I have raised my score.
> > > > > >
> > > > > > I still think that most baselines compared in the paper, including Drop-Upcycling and NVShard, are expected to show their advantages at larger training scales. With the relatively small training budget used here, these methods (including Sparse Upcycling) are not expected to perform well, making it difficult to draw strong conclusions about FineRMoE’s superiority. In this limited-training setting, the gains over the dense model are modest, while the MoE conversion still incurs inference overhead.
> > > > > >
> > > > > > I believe that including results at larger token budgets, as well as a more direct comparison of inference efficiency against the dense baseline, would be valuable for readers.

---

> ### Author Response · Authors · 2025-11-28
> **Response.**
>
> Thank you very much for your valuable suggestions and kind affirmation in raising the score to 6! Your comments are of great significance to the improvement of this work.
>
> Best Regards,
>
> The authors

---

### Official Review · Reviewer_j9xD · 2025-10-29

**Soundness:** 3
**Presentation:** 3
**Contribution:** 3
**Rating:** 6
**Confidence:** 3

**Summary:**

The paper presents FineRMoE, a fine-grained Mixture-of-Experts (MoE) architecture that decomposes the MLP block along both the intermediate and output dimensions.
In addition, the authors introduce a generalized upcycling method that extends beyond existing copy, allowing pretrained dense models to be converted into FineRMoE efficiently.
Extensive experiments on multiple Qwen2.5 model scales (0.5B–7B) demonstrate that the method consistently improves downstream performance.

**Strengths:**

1. High reproducibility: all training details, datasets, and hyperparameters are explicitly reported, increasing the paper’s credibility.
2. Robust empirical validation: experiments span multiple model sizes and show consistent gains.
3. Practical relevance: the proposed method can be readily applied to existing pretrained dense LLMs.
4. Experimental evidence: In the reported experiments, the model’s performance drops after CT, yet the proposed method achieves improvement, which strongly supports its effectiveness.

**Weaknesses:**

1. Missing comparison with Drop-Upcycling: Although cited as a related method, Drop-Upcycling is not included in Sec. 4.1 baseline comparisons. The omission leaves unclear whether FineRMoE’s gains hold against the strongest existing upcycling techniques.
2. Unclear effectiveness when CT does not degrade: In cases where CT does not cause performance drops (e.g., with weaker pretrained models such as Llama-3, or using higher quality datasets), it remains unclear whether FineRMoE would still outperform standard CT.
Demonstrating such results would clarify whether the gains arise from robustness to degradation or from improvements.

**Questions:**

1. Have you analyzed the model’s downstream performance when equal computational cost (not token count or FLOPs but GPU hours) is enforced during continual training (CT)? Conducting comparisons under the actual GPU time required for training would allow evaluation of the method’s usefulness while also accounting for potential differences in training throughput efficiency across model architectures.
(Using FLOPs alone does not necessarily ensure fairness between MoE and dense models, as their computational efficiency and hardware utilization characteristics differ.)

---

> ### Author Response · Authors · 2025-11-20
> **Response Part 1.**
>
> We are grateful for your pointing out the detailed problems and suggestions for modification. We hope the responses to each of your comments can address your concerns.
>
> - **Response of W1:**
> 1. Thank you for your valuable suggestion. Following the open-source code of Drop-Upcycling and its core function `shuffle_and_partially_initialize`, we implement Drop-Upcycling based on Qwen2.5 with sizes of 0.5B, 1.5B and 7B by setting its sampling ratio $r$ as 0.5. After training with 50B tokens in consistency with the experiments in Table 1, results are presented below.
>
> | size        | 0.5B  | 1.5B  | 7B    |
> | ----------- | ----- | ----- | ----- |
> | \#Experts   | 8     | 8     | 8     |
> | \#A-Experts | 2     | 2     | 2     |
> | \#Param/B   | 2.83  | 9.87  | 47.54 |
> | \#A-Param/B | 0.94  | 2.93  | 13.32 |
> | MMLU        | 24.36 | 25.93 | 25.71 |
> | BBH         | 6.93  | 10.17 | 17.98 |
> | HellaSwag   | 28.51 | 30.07 | 34.07 |
> | WinoGrande  | 50.67 | 52.41 | 51.85 |
> | ARC-C       | 21.93 | 20.99 | 25.51 |
> | ARC-E       | 40.70 | 43.27 | 51.30 |
> | AGIEval     | 25.86 | 25.52 | 25.65 |
> | MBPP        | 0.00  | 0.00  | 0.00  |
> | GSM8K       | 2.27  | 1.90  | 2.81  |
> | GPQA        | 25.00 | 24.78 | 23.66 |
> | Average     | 22.62 | 23.50 | 25.85 |
>
> 2. **Benchmark Performance.** In the paper of Drop-Upcycling, experiments are performed by training for 500B tokens. For a fair comparison with other methods, we follow the setting in Table 1 and limit training to 50B tokens. Under this condition, **Drop-Upcycling achieves the performance far inferior to that of FineRMoE**. We infer the reason as the existence of part of re-initialized parameters. Consequently, Drop-Upcycling begins with a higher training loss and converges more slowly. These results demonstrate that FineRMoE also exhibits **data-efficiency** compared with Drop-Upcycing in building MoE models from dense models.
> 3. **Parameter Efficiency.** Besides, the amount of total and activated parameters of Drop-Upcycling are both larger than FineRMoE, underscoring the superior parameter-efficiency of our method.
> 4. Furthermore, we also implement the method (arXiv:2410.07524), dubbed as **NVShard**, as another baseline based on Qwen2.5 for a comprehensive comparison. Please refer to **Response of W8 to Reviewer WeGh** for details. Results are also presented as below. FineRMoE achieves an average performance advantage of around 5 points than NVShard across three model sizes, demonstrating the effectiveness of our method. In addition, despite comparable activated parameter counts, NVShard contains about twice the total parameters of FineRMoE, underscoring FineRMoE's higher parameter efficiency. We have added the results of Drop-Upcycling and NVShard in the Sec. 4.1 and Table 1 for a more comprehensive baseline comparison. Besides, the inference efficiency of Drop-Upcycling and NVShard are added to the Appendix C and Table 6 for comparisons.
>
> | size        | 0.5B  | 1.5B  | 7B    |
> | ----------- | ----- | ----- | ----- |
> | \#Experts   | 64    | 64    | 64    |
> | \#A-Experts | 8     | 8     | 8     |
> | \#Param/B   | 2.83  | 9.88  | 47.55 |
> | \#A-Param/B | 0.63  | 1.78  | 7.63  |
> | MMLU        | 39.49 | 56.96 | 69.81 |
> | BBH         | 27.98 | 41.13 | 66.67 |
> | HellaSwag   | 49.44 | 66.72 | 79.05 |
> | WinoGrande  | 55.56 | 63.30 | 74.27 |
> | ARC-C       | 34.56 | 49.49 | 59.81 |
> | ARC-E       | 65.40 | 77.82 | 84.97 |
> | AGIEval     | 26.79 | 36.78 | 49.36 |
> | MBPP        | 16.80 | 33.80 | 53.20 |
> | GSM8K       | 26.84 | 64.67 | 83.62 |
> | GPQA        | 28.57 | 31.03 | 31.03 |
> | Average     | 37.14 | 52.14 | 65.18 |

---

> ### Author Response · Authors · 2025-11-20
> **Response Part 2.**
>
> - **Response of W2:**
> 1. Thank you for your valuable suggestion. Since the compared baselines construct MoE based on models like T5 (C32A2), LLaMA2-7B (S16A4), or Nemotron (arXiv:2410.07524), which possess relatively weak capabilities, achieving performance gains is relatively straightforward. In this work, we aim to validate the effectiveness of our method by employing a stronger model. Consequently, we adopt Qwen2.5, which currently represents the strongest base model in the upcycling literature.
> 2. Since we do not have other higher-quality data available, we instead use weaker pre-trained models for validation.
> 3. Within the Megatron-LM framework, conducting experiments with Llama-3 would require rebuilding the mmap for the whole dataset using its tokenizer, posing challenges to our experimental time and storage. Since Qwen2.5 and Qwen2 both use the Qwen2Tokenizer, we instead select the weaker Qwen2-1.5B model to conduct experiments for a quick validation.
> 4. Specifically, following the same data mixture strategy as in Table 1, we compare PT, CT, and FineRMoE after training on 10B tokens. Results presented below show that **the performance of dense CT has been improved compared with the pre-trained model**. Specifically, **FineRMoE consistently outperforms CT and achieves the best performance**. Combined with the results in Table 1, we may conclude that the performance gains are brought by the design of FineRMoE, verifying its effectiveness.
>
> | Qwen2-1.5B   | PT    | CT    | FineRMoE |
> | ------------ | ----- | ----- | -------- |
> | Train Tokens | N/A   | 10B   | 10B      |
> | MMLU         | 55.96 | 56.04 | 54.38    |
> | BBH          | 36.55 | 37.40 | 38.34    |
> | HellaSwag    | 66.94 | 67.78 | 66.70    |
> | WinoGrande   | 65.27 | 65.19 | 65.90    |
> | ARC-C        | 43.86 | 42.75 | 42.75    |
> | ARC-E        | 72.73 | 71.34 | 71.97    |
> | AGIEval      | 36.62 | 37.53 | 39.34    |
> | MBPP         | 37.80 | 33.80 | 39.40    |
> | GSM8K        | 46.90 | 53.37 | 55.65    |
> | GPQA         | 30.14 | 32.37 | 30.58    |
> | Average      | 49.28 | 49.76 | 50.50    |
>
> - **Response of Q1:**
>
> Thank you for your valuable suggestion. It is worth for deeply comparing the methods. Therefore, we perform additional experiments based on Qwen2.5-1.5B using **4 nodes (32 GPUs) and training for 16 hours** with the Megatron-LM framework. Specifically, for C32A2, S16A4 and FineRMoE, the parallelism are kept the same: TP=1, PP=2, CP=1, ETP=1, EP=8. For CT, the parallelism setting is TP=1, PP=1. The results are presented below. Under equal computational cost, FineRMoE outperforms the baselines and achieves the highest average performance. This further validates the usefulness of the proposed FineRMoE in continual training.
>
> | 32 GPUs, 16 Hours | PT    | CT    | C32A2 | S16A4 | FineRMoE |
> | ----------------- | ----- | ----- | ----- | ----- | -------- |
> | MMLU              | 60.87 | 60.63 | 61.01 | 25.94 | 59.61    |
> | BBH               | 43.33 | 45.09 | 45.40 | 21.10 | 46.34    |
> | HellaSwag         | 67.84 | 68.55 | 68.82 | 33.43 | 68.11    |
> | WinoGrande        | 64.88 | 65.19 | 65.59 | 52.25 | 65.11    |
> | ARC-C             | 54.86 | 53.67 | 53.58 | 24.15 | 52.99    |
> | ARC-E             | 81.02 | 80.77 | 80.81 | 46.21 | 80.51    |
> | AGIEval           | 39.83 | 38.91 | 38.85 | 25.75 | 40.83    |
> | MBPP              | 43.40 | 42.20 | 45.00 | 0.00  | 48.20    |
> | GSM8K             | 65.73 | 66.34 | 66.79 | 1.36  | 67.25    |
> | GPQA              | 32.14 | 31.70 | 30.58 | 22.10 | 32.14    |
> | Average           | 55.39 | 55.31 | 55.64 | 25.23 | 56.11    |

---

> ### Author Response · Authors · 2025-11-28
> **Response to Reviewer j9xD**
>
> Dear Reviewer j9xD,
>
> I hope this message finds you well. We sincerely appreciate the valuable suggestions you have provided, as they are extremely important for improving our work. We have provided a point-by-point response and analysis to each of the comments you raised. We look forward to your reply, and we would like to ensure whether we have fully addressed your concerns.
>
> Thank you again for the time and effort you have dedicated to reviewing our paper.
>
> Best Regards,
>
> the authors

---

### Official Review · Reviewer_p2SQ · 2025-11-01

**Soundness:** 3
**Presentation:** 3
**Contribution:** 1
**Rating:** 2
**Confidence:** 4

**Summary:**

This paper proposes a new architecture for MoE models, introducing a bi-level sparsity paradigm for the sparse experts. Specifically, in the second stage, it integrates experts using concatenation rather than the conventional summation approach. The paper claims this allows the information from each expert to be output without being mixed together. Since standard upcycling methods are not applicable for initializing this new architecture, a compatible upcycling approach is also developed. The experimental results are inconsistent; the proposed method sometimes outperforms and sometimes underperforms the baseline and common upcycling techniques.

**Strengths:**

To my knowledge, no well-known Sparse MoE (SMoE) models have adopted concatenation as an internal mechanism. The experimental results from this exploration could potentially aid future development in this area.

**Weaknesses:**

- The most important problem with this paper is that it doesn't discuss a strong necessity for introducing the additional structure (concat) into sparse experts. The paper describes the qualitative features of the proposed method (e.g., that concatenation allows different information to coexist without being mixed, unlike summation), but it fails to mention a specific situation that *must* be solved by the proposed method rather than by other implementations. This makes it difficult to distinguish whether the method was proposed to solve a genuine problem or simply because it was unexplored.
- I can’t find any significant improvements on the proposed method from the experimental results. They show only very similar results between the baseline and the proposed method by aggregating winner/loser subtasks, suggesting that the difference of average is still within the margin of error. As standard MoE (C32A2) marked similar results as well, I doubt that the training was actually insufficient to make meaningful comparison.

**Questions:**

Related to the weaknesses, please provide a more robust discussion comparing this to other MoE methods. As written above, it is not enough to distinguish the technical difference between theirs, but it is necessary to make some discussion about how especially the proposed method resolves something difficult in other methods.

---

> ### Author Response · Authors · 2025-11-20
> **Response Part 1.**
>
> Thanks for your comments. We hope the point-to-point responses could solve your problems.
>
> - **Response of W1:**
> 1. We begin by outlining our motivation. Models such as DeepSeek-V2 have demonstrated the effectiveness of fine-grained experts, leading to their increasing adoption. However, in these models, the input and output dimensions of experts remain significantly larger than their intermediate dimensions, resulting in persistent inter-expert parameter redundancy. We therefore investigate whether an even finer-grained design can enhance expert specialization to improve model performance.  This paper aims to explore such an innovative architecture, with the goal of contributing a solution for building MoE models that feature more highly specialized experts, stronger performance, and reduced parameter redundancy.
> 2. Motivated by this, we draw inspiration from multi-head attention to achieve finer-grained experts by compressing the output dimension of experts. However, this design leads to a dimension mismatch between the MoE layer's output and subsequent computations, following the standard weighted-sum fusion of in conventional MoEs. To resolve this, we propose a novel forward computation that first performs weighted summation of expert outputs in each group, followed by a sparse selection and concatenation of the outputs.
> 3. As detailed in the **Response of W2** below, the comprehensive evaluation across 10 general-domain benchmarks demonstrates that FineRMoE achieves the best performance and superior parameter efficiency compared to the baseline methods. Furthermore, FineRMoE exhibits significant higher inference efficiency. These results collectively underscore **the necessity and effectiveness of our method in holistically enhancing model generality, parameter efficiency, and inference efficiency jointly**.
>
> - **Response of W2:**
> 1. **Compared to PT, CT, and S16A4**, FineRMoE achieves substantial performance improvements across 10 benchmarks covering general, mathematical, and coding tasks. Specifically, FineRMoE outperforms PT and CT over 1 point on sizes of 1.5B and 7B. Moreover, FineRMoE attains an average **performance improvement of at least 70\%** over S16A4 across all three model sizes.
> 2. **Compared to C32A2**, FineRMoE not only outperforms it across 10 benchmarks, but also achieves this with **superior parameter and inference efficiency**. Specifically, it **owns fewer than $\frac{1}{6}$ of the total parameters of C32A2** across three model sizes. Furthermore, as shown in Table 6, FineRMoE's **TTFT** (178.3 ms) is **281 times faster** than that of C32A2 (50245.9 ms), and FineRMoE's **throughput** (27.3 tokens/s) is **136 times higher** than that of C32A2 (0.2 tokens/s).
> 3. In summary, FineRMoE demonstrates significant superiority over baseline methods on **benchmark performance, parameter efficiency, and inference efficiency**.
>
> - **Response of Q1:**
> 1. The key technical differences of FineRMoE are summarized as follows::
> * **Finer-grained expert:** We pioneer a finer-grained MoE architecture. Unlike existing experts that only compress the intermediate dimension, FineRMoE compresses both the intermediate and output dimensions to enhance expert specialization and mitigate redundancy.
> * **Novel forward computation:** We design a novel computation paradigm for the MoE layer. Unlike the standard weighted sum used in conventional single-layer MoEs, FineRMoE introduces two distinct sparse layers, i.e., a sparse sum layer and a sparse concatenation layer, for multi-expert fusion.
> * **Unified router mechanism:** We introduce a novel router mechanism that utilizes a single router for activation in both two sparse layers, eliminating the computational overhead associated with using two separate routers.
> * **Generalized upcycling method:** We propose a generalized upcycling method. Unlike existing methods tailored for specific architectures, our approach is generalizable. It is applicable to FineRMoE with its two-layer design, single-layer MoEs with experts replicated from the FFN, and single-layer MoEs with experts split from the FFN along the intermediate dimension, thereby addressing the generality limitation in prior work.
> 2. Through these designs, according to the **Response of W2**, FineRMoE demonstrates significant improvements over baseline methods in addressing the **three key challenges** jointly: enhancing model performance, designing parameter-efficient architectures, and improving inference efficiency.

---

> ### Author Response · Authors · 2025-11-28
> **Response to Reviewer p2SQ**
>
> Dear Reviewer p2SQ,
>
> I hope this message finds you well. We sincerely appreciate the valuable suggestions you have provided, as they are extremely important for improving our work. We have provided a point-by-point response and analysis to each of the comments you raised. We look forward to your reply, and we would like to ensure whether we have fully addressed your concerns.
>
> Thank you again for the time and effort you have dedicated to reviewing our paper.
>
> Best Regards,
>
> the authors

---

### Official Review · Reviewer_oeAo · 2025-11-02

**Soundness:** 3
**Presentation:** 2
**Contribution:** 3
**Rating:** 6
**Confidence:** 4

**Summary:**

This paper extends fine-grained expert design in MoE models from the intermediate dimension to the output dimension. The proposed method adopts a bi-level sparsity paradigm: a sparse sum layer generates dimension-reduced candidate vectors via sparsely activated fine-grained experts, while a sparse concatenation layer restores the output dimension by selectively concatenating these candidates, with a single router network controlling both expert activation and candidate selection to avoid dual-router overhead. Experimentally, the proposed method built on Qwen2.5 (0.5B, 1.5B, 7B) with 128 experts (2 activated per token) via this upcycling method, trained on 50B tokens, outperforms baselines across 10 benchmarks in performance, parameter efficiency, and inference efficiency.

**Strengths:**

* Extended Fine-Grained Design: the proposed method innovatively extends fine-grained expert design from the intermediate dimension to the output dimension of MoE models, addressing the long-standing issue of dimensional inconsistency that limited output-dimension specialization in previous MoEs, thus enhancing expert redundancy reduction and specialization.
* Generalized Upcycling Method: The proposed upcycling approach resolves incompatibilities between existing upcycling techniques (for single-layer, weighted-sum MoEs). It efficiently initializes shared and sparse experts using pre-trained FFN weights (via copying or splitting) and remains compatible with mainstream upcycling methods (e.g., FFN replication), reducing training costs significantly.
* Superior Performance & Efficiency: Experiments on Qwen2.5 (0.5B, 1.5B, 7B) show the proposed method outperforms baselines (dense models, other MoEs like C32A2) across 10 benchmarks in average performance, while achieving better parameter efficiency (outperforming larger-parameter MoEs) and inference efficiency (lower prefill latency, higher decoding throughput).

**Weaknesses:**

* The proposed method’s performance depends on the proper tuning of four hyperparameters. Improper configurations may lead to suboptimal expert specialization or increased computational overhead, adding complexity to model deployment.
* This paper lacks experiments to validate the effectiveness of FineRMoE in Reinforcement Learning scenarios. Throughout the experimental sections, the evaluations are exclusively conducted on ten standard benchmarks covering knowledge, reasoning, code, and math, with no design or results of experiments involving RL tasks. As a result, the adaptability and performance of the FineRMoE architecture and its upcycling method in RL-related applications remain unproven.

**Questions:**

* Could you provide more specific details regarding the load balancing loss?
* Why does the continued training (CT) dense model exhibit slightly worse performance compared to the pre-trained model?
* Furthermore, could the CT model’s performance be improved with additional training data—and if so, would this also affect the performance gap between the MoE and FineRMoE models?

---

> ### Author Response · Authors · 2025-11-20
> **Response Part 1.**
>
> Thank you for your careful review on this paper. We hope the responses to each of your comments in the following could ease your concerns.
>
> - **Response of W1:**
> 1. According to the analysis in Sec. 4.4 (Table 3) and Appendix F (Table 8), except for the first four fine-grained settings in Table 3, all other settings in that table and all configurations in Table 8 achieve better performance than the pre-trained model. Therefore, our method requires no specific hyperparameter tuning and can be readily deployed to improve performance.
> 2. Moreover, this work aims to achieve a finer-grained expert design compared to existing MoE architectures. According to the analysis in Sec. 4.4 (Table 3), larger settings of $G_I$ and $G_O$ generally lead to stronger model performance and relatively fewer total/activated parameters. Therefore, in practice, the fine-grained hyperparameters can be set to relatively higher values to enhance parameter efficiency and overall performance.
>
> - **Response of W2:**
> 1. We sincerely appreciate this valuable suggestion. Validating FineRMoE in reinforcement learning scenarios represents a promising direction, and we will explore it in our future work.
> 2. Currently, the existing literature on upcycling discussed in our related work has not incorporated RL training into their validation experiments. Therefore, it is challenging to find suitable baselines for comparison. For this reason, our study focuses exclusively on continual training to compare methods and analyze their advantages.
>
> - **Response of Q1:**
>
> Thank you for your advice. In detail, we follow DeepSeek-V2 for the load balancing loss design. Given the number of experts as $N$, the number of activated experts per token as $G_OT_I$, the number of tokens in a sequence as $L$, the score of assigning the $t$-th token to the $i$-th expert as $s_{i,t}$, the load balancing loss $L_{lbl}$ is calculated as:
> $$\begin{aligned}
> L_{lbl} = \alpha\sum_{i=1}^{N}f_iP_i, \\
> P_i =\frac{1}{L}\sum_{t=1}^{L}s_{i,t}, \\
> f_i = \frac{N}{G_OT_IL} \sum_{t=1}^{L}\mathbf{I}_{i,t},
> \end{aligned}$$
>
> where $\mathbf{I}_{i,t}$ is an indicator function, which equals to 1 when the $t$-th token selects the $i$-th expert, otherwise 0. Specifically, the FineRMoE is optimized with the sum of language modeling loss and the weighted load balancing loss, and the weight of load balancing loss $\alpha$ is set as 0.001 throughout our experiments. We have added this demonstration in Appendix B in the updated PDF.
>
> - **Response of Q2:**
> 1. From the data perspective, the discrepancy between continual training data and pre-training data results in performance improvements on some benchmarks while leading to degradation on others.
> 2. For the dense model, all parameters are activated during both training and inference, meaning that continual pre-training with new data affects the entire model. This often leads to catastrophic interference as new knowledge conflicts with previously acquired knowledge. For instance, concerning the models built on Qwen2.5-1.5B, Continual Training (CT) improves performance on benchmarks such as BBH, HellaSwag, and GSM8Km, but leads to degradation on ARC-C, ARC-E, and MBPP. Consequently, the overall performance declines when the degradation outweighs the gains.
> 3. In contrast, in MoE models, only a subset of experts is activated for each input. This sparse activation enables the model to acquire new knowledge efficiently while preserving its pre-trained capabilities. As a result, MoE models achieve more comprehensive performance gains with fewer conflicts compared to continually trained (CT) dense models.
> 4. We have added this explanation in the Sec. 4.1 in the updated PDF.

---

> ### Author Response · Authors · 2025-11-20
> **Response Part 2.**
>
> - **Response of Q3:**
> 1. As indicated by the first two points in the **Response of Q2**, leveraging extra high-quality data has the potential to enhance the overall performance of CT. Data quality plays a vital role in both pre-training and continual training.
> 2. The results reported in Table 1 are achieved using the optimal data mixture with 50B tokens trained, identified through data ablation studies on our available datasets. To answer the question that **"could the CT model’s performance be improved with additional training data"**, we conduct 3 sets of experiments: training FineRMoE and CT under the optimal data mixture with 10B tokens, and another two experiments trained on 10B tokens with inferior data mixtures. The results based on Qwen2.5-1.5B are presented below. They demonstrate that (i) under the same optimal data mixture, increasing the training data from 10B to 50B tokens, the performance of CT and FineRMoE can be improved; (ii) under the same amount of training data as 10B tokens, by improving the data mixture from DataMix1 and DataMix2 to the optimal one, the performance of both CT and FineRMoE can also been improved.
>
> |              | PT    | CT      | FineRMoE | CT      | FineRMoE | CT       | FineRMoE | CT       | FineRMoE |
> | ------------ | ----- | ------- | -------- | ------- | -------- | -------- | -------- | -------- | -------- |
> | Data Setting | N/A   | Optimal | Optimal  | Optimal | Optimal  | DataMix1 | DataMix1 | DataMix2 | DataMix2 |
> | Train Tokens | N/A   | 50B     | 50B      | 10B     | 10B      | 10B      | 10B      | 10B      | 10B      |
> | MMLU         | 60.87 | 60.70   | 59.64    | 60.38   | 59.30    | 60.48    | 59.15    | 60.60    | 59.42    |
> | BBH          | 43.33 | 45.15   | 47.09    | 44.22   | 45.97    | 43.94    | 45.29    | 43.51    | 45.51    |
> | HellaSwag    | 67.84 | 68.68   | 68.17    | 68.10   | 67.51    | 67.66    | 67.18    | 67.40    | 67.11    |
> | WinoGrande   | 64.88 | 65.43   | 65.43    | 64.09   | 65.27    | 65.27    | 64.40    | 64.40    | 65.11    |
> | ARC-C        | 54.86 | 53.41   | 53.33    | 52.73   | 53.41    | 51.71    | 51.62    | 52.22    | 52.65    |
> | ARC-E        | 81.02 | 80.72   | 80.81    | 80.64   | 80.35    | 79.92    | 80.43    | 80.13    | 80.64    |
> | AGIEval      | 39.83 | 38.91   | 43.14    | 39.73   | 41.81    | 38.73    | 41.28    | 38.53    | 41.09    |
> | MBPP         | 43.40 | 41.20   | 50.40    | 43.40   | 50.00    | 41.60    | 47.20    | 40.60    | 46.00    |
> | GSM8K        | 65.73 | 66.11   | 65.81    | 66.49   | 66.34    | 67.48    | 68.39    | 67.93    | 67.63    |
> | GPQA         | 32.14 | 31.47   | 32.37    | 31.70   | 32.14    | 30.36    | 32.14    | 31.47    | 30.58    |
> | Average      | 55.39 | 55.18   | 56.62    | 55.15   | 56.21    | 54.72    | 55.71    | 54.68    | 55.57    |
>
> 3. In addition, under the same data configuration, whether the CT model’s performance can be improved also depends on the capability of the base model. The performance of CT and FineRMoE, when built on an inferior base model, can surpasss that of the pre-trained model. Specifically, under the same data mixture strategy as in Table 1, we compare the results of PT, CT and FineRMoE built on Qwen2-1.5B with 10B tokens trained. Results are presented below. They show that for Qwen2-1.5B, after continual training on 10B tokens, the average performance of CT (49.76) and FineRMoE (50.50) both surpasses that of the pre-trained model (49.28).
>
> | Qwen2-1.5B   | PT    | CT    | FineRMoE |
> | ------------ | ----- | ----- | -------- |
> | Train Tokens | N/A   | 10B   | 10B      |
> | MMLU         | 55.96 | 56.04 | 54.38    |
> | BBH          | 36.55 | 37.40 | 38.34    |
> | HellaSwag    | 66.94 | 67.78 | 66.70    |
> | WinoGrande   | 65.27 | 65.19 | 65.90    |
> | ARC-C        | 43.86 | 42.75 | 42.75    |
> | ARC-E        | 72.73 | 71.34 | 71.97    |
> | AGIEval      | 36.62 | 37.53 | 39.34    |
> | MBPP         | 37.80 | 33.80 | 39.40    |
> | GSM8K        | 46.90 | 53.37 | 55.65    |
> | GPQA         | 30.14 | 32.37 | 30.58    |
> | Average      | 49.28 | 49.76 | 50.50    |

---

> ### Author Response · Authors · 2025-11-20
> **Response Part 3.**
>
> - **Response of Q3:**
> 4. To answer the question that **"would this also affect the performance gap between the MoE and FineRMoE models?"**, we conduct another set of experiments using DataMix1 based on Qwen2.5-1.5B. The CT, C32A2, S16A4 and FineRMoE (ours) are trained on 10B tokens for a quick validation. Results are presented below.（i）According to the results, in consistent with Table 1, FineRMoE still outperforms other MoE architectures, i.e. C32A2 and S16A4, **maintaining a stable performance gap**, demonstrating the consistent advantage of FineRMoE under different data mixtures and training data sizes. (ii) Compared with the results in Table 1, by improving the data mixture from DataMix1 to the optimal one and increasing the training tokens from 10B to 50B tokens, the performance of CT, C32A2, S16A4, and FineRMoE can be uniformly improved.
>
> |              | PT    | CT       | C32A2    | S16A4    | FineRMoE |
> | ------------ | ----- | -------- | -------- | -------- | -------- |
> | Data Setting | N/A   | DataMix1 | DataMix1 | DataMix1 | DataMix1 |
> | Train Tokens | N/A   | 10B      | 10B      | 10B      | 10B      |
> | MMLU         | 60.87 | 60.48    | 60.70    | 25.07    | 59.15    |
> | BBH          | 43.33 | 43.94    | 44.19    | 9.08     | 45.29    |
> | HellaSwag    | 67.84 | 67.66    | 67.57    | 28.94    | 67.18    |
> | WinoGrande   | 64.88 | 65.27    | 65.75    | 49.49    | 64.40    |
> | ARC-C        | 54.86 | 51.71    | 51.37    | 20.56    | 51.62    |
> | ARC-E        | 81.02 | 79.92    | 79.42    | 35.19    | 80.43    |
> | AGIEval      | 39.83 | 38.73    | 38.76    | 25.57    | 41.28    |
> | MBPP         | 43.40 | 41.60    | 41.40    | 0.00     | 47.20    |
> | GSM8K        | 65.73 | 67.48    | 67.85    | 1.74     | 68.39    |
> | GPQA         | 32.14 | 30.36    | 30.58    | 22.10    | 32.14    |
> | Average      | 55.39 | 54.72    | 54.76    | 21.77    | 55.71    |

---

> ### Author Response · Authors · 2025-11-28
> **Response to Reviewer oeAo**
>
> Dear Reviewer oeAo,
>
> I hope this message finds you well. We sincerely appreciate the valuable suggestions you have provided, as they are extremely important for improving our work. We have provided a point-by-point response and analysis to each of the comments you raised. We look forward to your reply, and we would like to ensure whether we have fully addressed your concerns.
>
> Thank you again for the time and effort you have dedicated to reviewing our paper.
>
> Best Regards,
>
> the authors

---

### Author Response · Authors · 2025-11-24
**General Response**

Dear Area Chair and Reviewers,

Firstly, we greatly appreciate the constructive comments and valuable suggestions for our work. They are precious for improving our work and the quality of the paper. Thanks the reviewers for approving our **novel idea in finer-grained expert design and generalized upcycling method, superior performance and efficiency, high reproducibility, clear written and strong practical advantage**. To fully address the concerns raised by the reviewers, we have conducted **additional experiments and analysis** during the rebuttal period.

**Experiments**

**For Reviewer oeAo (initial score: 6)**
- We conduct a series of experiments by using different data mixture strategies and training tokens to verify the **stable performance advantage of the proposed FineRMoE over baseline methods**.

**For Reviewer j9xD (initial score: 6)**
- We perform additional experiments on **Drop-Upcycling and NVShard**, which are added as new baseline methods, to verify **the effectiveness of the proposed FineRMoE compared with more methods**.
- We conduct experiments by employing weaker model (Qwen2-1.5B) to validate **the effectiveness of the proposed FineRMoE when applying to different models**.
- We perform a set of experiments by training under limited equal computation budget to verify **the cost-effectiveness of the proposed FineRMoE that achieves the best performance with the same training cost**.

**For Reviewer WeGh (initial score: 4, then raised to 6)**
- We perform evaluation of the resulted models in Table 1 on more benchmarks to validate **the effectiveness of the proposed FineRMoE against baseline methods on more evaluation**.
- We conduct experiments by employing stronger model (Qwen3-0.6B) to validate **the effectiveness of the proposed FineRMoE when applying to different models**.
- We perform additional experiments on **Drop-Upcycling and NVShard**, which are added as new baseline methods, to verify **the effectiveness of the proposed FineRMoE compared with more methods**.
-  We perform a training from scratch to show the **potential of the FineRMoE in building MoE models from scratch**.
- We conduct an ablation study on the router design to prove that **the proposed routing mechanism with a single router is more effective than using separate routers**.

**Detailed demonstrations**

In addition to the experiments, we also supplied more **detailed demonstrations** of our method, including:
- For Reviewer oeAo and WeGh, we explain that our method **does not require extensive hyper-parameter tuning**, most of the settings could achieve performance gains, and finer-grained experts can lead to better performance and fewer parameters, which is also the goal of this work.
-  For Reviewer oeAo and WeGh, we provide the detailed setting of the **load balancing loss, training data, warmup steps and the Megatron-LM version**.
- For Reviewer p2SQ and WeGh, we highlight the performance advantage that FineRMoE outperforms at least **70% over S16A4** across all three model sizes. The total parameter of C32A2 is **6 times more than** that of FineRMoE. While FineRMoE's TTFT (178.3 ms) is **281 times faster than that of C32A2** (50245.9 ms), FineRMoE's throughput (27.3 tokens/s) is **136 times higher than that of C32A2** (0.2 tokens/s).
- For Reviewer WeGh, we demonstrate the **technical differences** of our work, including: finer-grained expert, novel forward computation, unified router mechanism and generalized upcycling method.

For each reviewer, we provide the point-to-point response as below, and we also update the entire PDF. We are delighted to have a **detailed and in-depth discussion with Reviewer WeGh, addressing the concerns raised and ultimately achieving a positive score increase, leading to 3 positive ratings (6,6,6) out of 4**. We sincerely appreciate for your valuable comments and support.

Best Regards,

The authors

---

### Meta-Review · Area_Chair_GM9R · 2025-12-19

**Summary:**

This paper proposes FineRMOE extending fine-grained expert design in MoE models from the intermediate dimension to the output dimension. The work builds upon bi-level sparsity paradigm for the sparse experts. Compared conventional Integration stage where summation is leveraged, FineRMOE integrates experts using concatenation with a single router network controlling both expert activation and candidate selection to avoid dual-router overhead. The paper claims this allows the information from each expert to be output without being mixed together. Experimentally, the proposed method built on Qwen2.5 (0.5B, 1.5B, 7B) with 128 experts (2 activated per token) via this upcycling method, trained on 50B tokens, outperforms baselines across 10 benchmarks in performance, parameter efficiency, and inference efficiency.

Reviewers found the paper is of following strength

* Study a new approach (concatenation) to combine MoE outputs
* Working on a hot area (LLMs) and proposed method can be readily applied to existing LLMs.
* The experiments were conducted with reasonable benchmarks to demonstrate the effectiveness of proposed method. The evaluation setup is proper and makes the results convincing.
* The paper is very clear and well-written. Good reproducibility.

**Reviewer Concerns:**

At the beginning of the discussion, the reviewers had some questions such as consistency in performance gain, whether the gain is transferrable to reinforcement learning, dependency on hyperparameter setting, and insights about the method design.

**Reviewer Scores:**

Reviewers found the additional explanation from authors during rebuttal are helpful and explaining most of the questions. The additional insights, discussion, and experiment results definitely strengthen this work.

Although this submission is of reasonable quality and contribution to science community, the reviewers still found this work is at the borderline compared to a typical ICLR paper. We can see some gains from the proposed method on a series of LLMs. However, we still don't have enough evidence about the benefit and necessity of FineRMOE. E.g., the benefit of concatenation such as preserving different information is discussed qualitatively. We would love to see more and consistent ablation results on that. Due to this hesitation, we found it could be too early to introduce this paper to ICLR community and we encourage the authors to resubmit to a later conference.

---

### Decision · Program_Chairs · 2026-01-26

Reject